# Direct synthesis of oxalic acid via oxidative CO coupling mediated by a dinuclear hydroxycarbonylcobalt(III) complex

Yingzhuang Xu[1], Songyi Li[1] & Huayi Fang [1]✉

Oxidative coupling of CO is a straightforward and economic benign synthetic route for value-added $\alpha$-diketone moiety containing $C_2$ or higher carbon compounds in both laboratory and industry, but is still undeveloped to date. In this work, a rare coplanar dinuclear hydroxycarbonylcobalt(III) complex, bearing a Schiff-base macrocyclic equatorial ligand and a $\mu$-$\kappa^1$(O):$\kappa^1$(O')-acetate bridging axial ligand, is synthesized and characterized. The Co(III)-COOH bonds in this complex can be feasibly photocleaved, leading to the formation of oxalic acid. Moreover, the light-promoted catalytic direct production of oxalic acid from CO and $H_2O$ using $O_2$ as the oxidant with good selectivity (> 95%) and atom economy at ambient temperature and gas pressure based on this dicobalt(III) complex have been achieved, with a turnover number of 38.5. The [13]C-labelling and [18]O-labelling experiments confirm that CO and $H_2O$ act as the sources of the -COOH groups in the dinuclear hydroxycarbonylcobalt(III) complex and the oxalic acid product.

Selective C-C coupling of carbon monoxide (CO), which is one of the central $C_1$ feedstocks in both laboratory and chemical industries[1–3], is long known as an important and efficient synthetic route for $C_2$ and higher carbon products[4–7]. The majority of previously documented CO coupling strategies could be sorted into two categories as oxidative CO coupling and reductive CO coupling. Researches on reductive CO coupling can be traced back to the early nineteenth century during which molten potassium is reported to reductively couple CO to form the $[C_2O_2]_n^{2n-}$ anions[8,9]. Since then, a series of s-,[7] p-,[10,11] d-,[12–21] and f-block[22–24] element complexes have been demonstrated to be capable for the reductive CO coupling, affording a number of C-C bond formation products including ynediolates, enediolates, and oxygen-free hydrocarbons. In comparison, the research of oxidative CO coupling is much lagged behind and only a limited number of examples have been reported till now[25]. For instances, organolithiums are known to react with CO to afford 1,2-diketones as the oxidative CO coupling products[26–28]. Besides, some transition metal complexes, such as Re(I/III) and Pd(II) complexes, are also illustrated to be capable of mediating both stoichiometric and catalytic CO oxidative couplings[29–35] (Fig. 1a, route i). High pressure of CO (65–80 atm) is noted to be crucial for

good yields and selectivity of the reported catalytic systems, and carbonates are usually found to form as major products at low CO pressure in this processes34. Mechanistically, the high gas pressure is a prerequisite for the single site pathway, in which two CO molecules are activated at one metal center, that governs the aforementioned processes. Therefore, the cooperative mechanism standing on two or more geometrically correlated reacting sites potentially paves the way for oxidative CO coupling at ambient conditions[36].

Oxalic acid plays indispensable roles in various industry processes, such as metal processing, rare earth extraction, leather treatment and pharmaceutical, with an annual market of 350,000 tons[37]. And CO is one of the major raw materials used in the applied commercially viable production of oxalic acid, wherein CO is firstly converted to the alkali formate intermediate and the oxalic acid is obtained by the following formate coupling and acidification (Fig. 1a, route ii). However, harsh conditions, including high reaction temperature and CO pressure, are necessary for this multistep time- and energy-consuming route. Meanwhile, the formation of carbonates byproducts is inevitable as a consequence of the decompositions of the alkali formates and oxalate product. In principle, oxalic acid can be

[1]School of Materials Science and Engineering, Tianjin Key Lab for Rare Earth Materials and Applications, Nankai University, Tianjin 300350, China.
✉e-mail: hfang@nankai.edu.cn

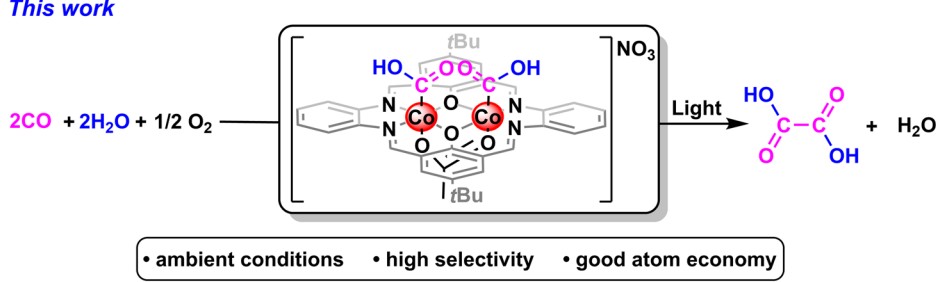

**a Reported synthetic routes of oxalic acid from CO**

**b Direct synthesis of oxalic acid via oxidative CO coupling**

*This work*

**Fig. 1 | Synthetic routes of oxalic acid from CO. a** Reported synthetic routes of oxalic acid from CO. **b** Direct synthesis of oxalic acid via oxidative CO coupling.

straightforwardly prepared by the oxidative coupling of CO with $H_2O$, while to the best of our knowledge, no such process has been reported yet.

Here, we show the light-promoted direct and selective production of oxalic acid from CO and $H_2O$ with good atom economy mediated by a coplanar dinuclear hydroxycarbonylcobalt(III) complex using $O_2$ as the oxidant under ambient reaction temperature (room temperature) and gas pressure (1 atm of CO and 1 atm of $O_2$) with a turnover number of 38.5.

## Results

### Synthesis and characterizations of dicobalt complexes 1-5

The macrocyclic ligand (**H₂L**) having two coplanar metal binding sites is synthesized based on the previously reported method[38]. The reaction of **H₂L** with 2 equivalents of $Co(OAc)_2$ under $N_2$ in ethanol followed by recrystallization with $Et_2O$ gives the isolation of yellow complex **1** in 86% yield (Fig. 2a). The solid-state structure of **1** shows that the two Co(II) centers sit at different sides of the ligand, with extra coordinations from acetate ligands (Fig. 3a). The two secondary benzylamine moieties in the ligand framework are converted to imines during the metalation, probably via the cobalt mediated dehydrogenation[39]. The $^1H$ NMR monitoring of the synthesis of **1** in a J. Young NMR tube showed a singlet resonance at $\delta = 4.35$ ppm, indicating the formation of $H_2$ (Supplementary Fig. 1). The effective magnetic moment (7.67 $\mu_B$) measured for **1** at room temperature is indicative of the presence of two high-spin ($S = 3/2$) Co(II) centers (Supplementary Fig. 2). The spin-orbital splitting energy (15.9 eV) obtained from X-ray photoelectron spectroscopy (*XPS*) measurement is also in agreement with the oxidation state assignments of +2 for the two cobalt centers (Supplementary Fig. 3). **1** can be readily oxidized by $O_2$ in methanol at room temperature to provide a red mixed-valent dicobalt(II/III) complex **2**, as supported by the *XPS* measurement (the measured spin-orbital splitting energies for Co(II) and Co(III) are 15.6 eV and 14.8 eV, respectively, Supplementary Fig. 5), in 81% isolated yield (Fig. 2b). The solid state structure of **2** shows that the Co(III) center well accommodates in the $N_2O_2$ plane

and occupies an octahedral coordination geometry, while the larger Co(II) center still sits above the ligand framework (Fig. 3b). In accordance, the Co(III)-$O_{equatorial}$ (1.881(3)/1.868(2) Å) and Co(III)-$N_{equatorial}$ (1.867(3)/1.870(3) Å) bonds in **2** are significantly shorter than those in **1** (Co(II)-$O_{equatorial}$: 2.063(2)/2.144(3) Å; Co(II)-$N_{equatorial}$: 2.108(3)/2.071(3) Å). The effective magnetic moment (4.12 $\mu_B$) of **2** indicates a high spin ($S = 3/2$) Co(II) center and a low spin ($S = 0$) Co(III) center (Supplementary Fig. 6).

After the addition of 1.1 equivalent of $Ce(NH_4)_2(NO_3)_6$, the brown dicobalt(III) complex **3** is obtained via the further oxidation of **2** at elevated reaction temperature in an isolated yield of 86% (Fig. 2c). The measured spin-orbital splitting energy (15.1 eV) is in line with the presence of only Co(III) centers (Supplementary Fig. 7). $^1H$ NMR spectrum of **3** displays sharp resonances in the range of $\delta = 1 – 10$ ppm, showing the diamagnetic feature of **3** (Supplementary Fig. 10). As shown in Fig. 3c, both of the Co(III) centers in **3** well fit into the $N_2O_2$ plane of the ligand. The Co(III)-$O_{equatorial}$ (1.914(2)/1.894(2)/1.896(2)/1.900(2) Å) and Co(III)-$N_{equatorial}$ (1.859(3)/1.874(3)/1.868(3)/1.866(3) Å) bond lengths are very comparable to those for the Co(III) center in **2**. The two Co(III) centers are bridged by an acetate ligand in a $\mu\text{-}\kappa^1(O)\text{:}\kappa^1(O')$ manner, while the hydroxyl and methoxyl axial ligands are coordinated to each of the Co(III) centers from the other side.

Treatment of **3** in methanol (with the presence of 0.01% *w/w* of water) with 1 atm of CO at 50 °C produces the diamagnetic dinuclear hydroxycarbonylcobalt(III) complex (**4**) within 24 h (Fig. 2d). The spin-orbital splitting energy (15.0 eV) derived from *XPS* measurement is nearly identical to that of **3** (Supplementary Fig. 12). Besides the different axial ligands, the molecular structure of **4** largely resemble the geometrical features of **3** (Fig. 3d). The C = O (1.193(5)/1.207(4) Å) and C(O)-OH (1.231(5)/1.249(4) Å) bond lengths are in the normal range reported for other hydroxylcarbonyl metal complexes[40,41]. The IR spectrum of **4** shows two close absorptions at 1697 and 1670 cm$^{-1}$, in satisfactory agreement with the modelling results based on density functional theory (DFT) calculations (1724 and 1685 cm$^{-1}$). When $^{13}CO$ is used in the synthesis of **4**, the two absorptions are red-shifted to 1662 and 1651 cm$^{-1}$ (Supplementary Fig. 13, the calculated values are 1685

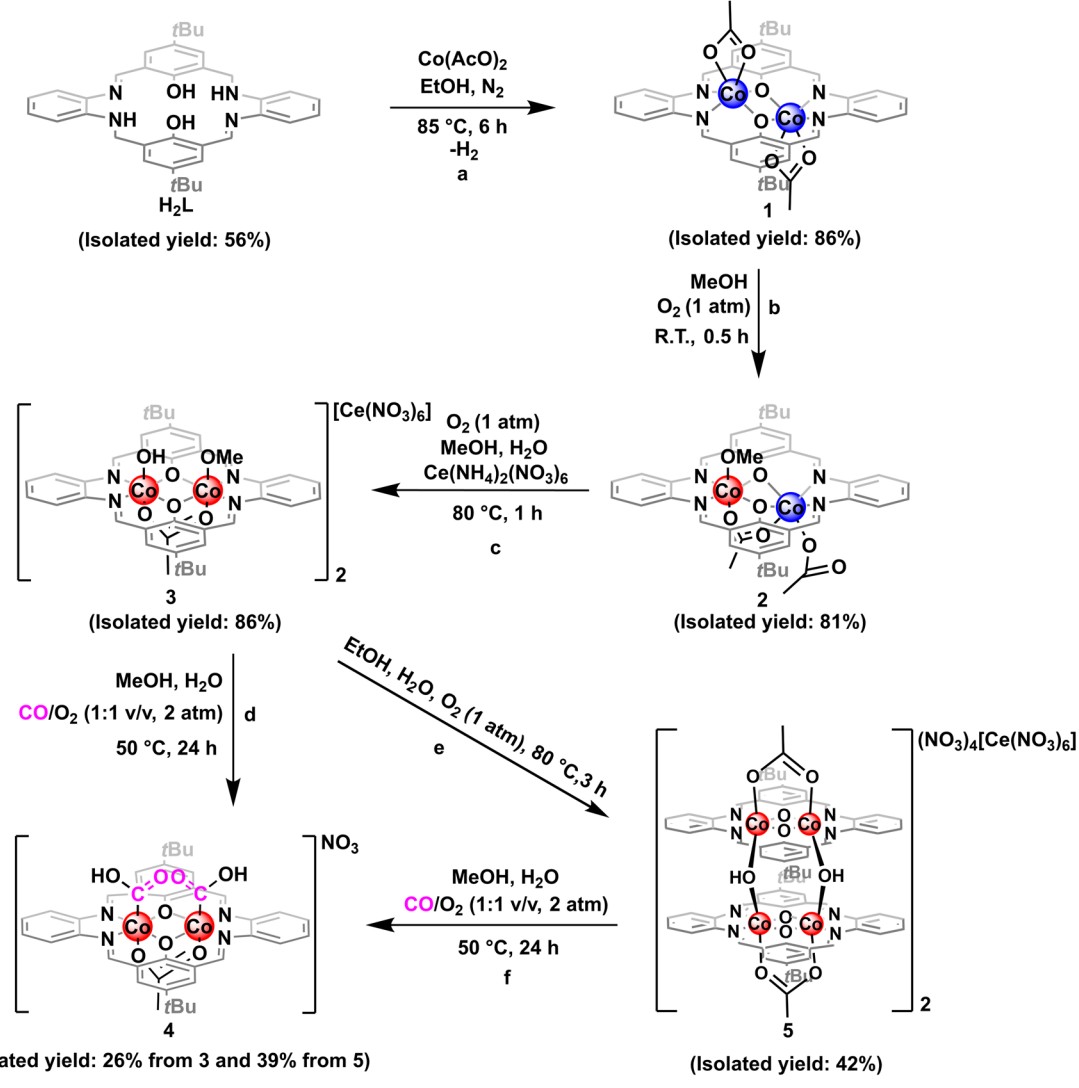

**Fig. 2 | Procedures for the synthesis of key complexes. a** Synthesis of **1**. **b** Synthesis of **2**. **c** Synthesis of **3**. **d** Synthesis of **4** from **3**. **e** Synthesis of **5**. **f** Synthesis of **4** from **5**.

and 1648 cm$^{-1}$). $^1$H NMR signals of the two carboxylic protons are found at $\delta = 13.34$ and 13.08 ppm, respectively (Supplementary Fig. 14). In addition, the thermogravimetric analysis of **4** is performed and a weight loss (10.1 $wt\%$) at 280−290 °C, as a result of the release of two -COOH groups, is observed (Supplementary Fig. 17). **4** is envisioned to form via the CO insertion into the Co(III)-OH bonds of the proposed dinuclear hydroxocobalt(III) intermediate, which could be generated via the reaction of **3** with H$_2$O that results in the replacement of axial methoxyl ligand to hydroxyl ligand. Although attempts for the isolation of this intermediate are unsuccessful, tetranuclear cobalt complex **5** is obtained in 42% isolated yield by heating the ethanol solution of **3** at 80 °C for 3 hours as shown in Fig. 2e. The measured spin-orbital splitting energy of cobalt centers in **5** (15.0 eV) is nearly identical to those of Co(III) centers in **3** and **4** (Supplementary Fig. 18). The lengths of the Co-O$_{\mu\text{-OH}}$ bonds (1.905(4)/1.909(4)/1.907(4)/1.895(4) Å, Fig. 4a) in **5** are very comparable to the reported values (1.888 − 1.912 Å) for dinuclear cobalt(III) μ-hydroxo species[40,41], but is shorter than the typical Co(III)-OH$_2$ bond (1.945 Å)[42,43] and significantly longer than the Co(III)-O$_{\mu\text{-O}}$ bonds (1.783−1.796 Å)[44]. The effective magnetic moment (1.14 $\mu_B$) measured for **5** at room temperature is indicative of the presence of only one unpaired electron (Supplementary Fig. 19). The unrestricted corresponding orbital analysis and the calculated spin density of **5** show that the unpaired electron density is on the cobalt centers (Supplementary Fig. 20). The EPR measurement of **5** in solid

state is also conducted at 97 K. An anisotropic signal ($g_1 = 2.023$, $g_2 = 2.222$, $g_3 = 2.305$) with well resolved hyperfine splitting from Co nucleus ($I = 7/2$, $A_1 = 264.00$ MHz, $A_2 = 64.50$ MHz, $A_3 = 60.00$ MHz) is observed (Fig. 4b). Furthermore, **4** can be obtained by the reaction of **5** with CO and O$_2$ (Fig. 2f). A plausible formation pathway of **5** from **3** via the proposed dinuclear hydroxylcobalt(III) intermediate is depicted in Supplementary Fig. 21.

### Light promoted production of oxalic acid catalyzed by 4

The relatively short distance (3.419 Å) between the two carbon atoms of the -COOH ligands in **4** encourages us to further examine the production of oxalic acid from **4**. Irradiation of **4** using Xe-lamp as the light source is carried out under N$_2$ at room temperature, and oxalic acid is observed to form in 57% yield. By replacing the N$_2$ atmosphere to CO/O$_2$ mixture gas (1:1 $v/v$, 2 atm), the catalytic production of oxalic acid with good selectivity (> 95%) is achieved with a turnover number (TON) of 38.5 (Table 1). Only trace amount (TON ~ 0.3) of dimethyl carbonate, which is a commonly seen side product during the Pd(II) complexes catalyzed production of oxalic acid[31,34,35], is formed, and, meanwhile, no dimethyl oxalate is found in this process. It is worth noting that the formation of H$_2$O$_2$ (TON ~ 0.1) is also detected, which might indicate the intermediacy of the dinuclear hydroxocobalt(III) complex in the catalytic cycle. It is noteworthy that yellow solid was observed to precipitated out of the reaction solution during the catalytic

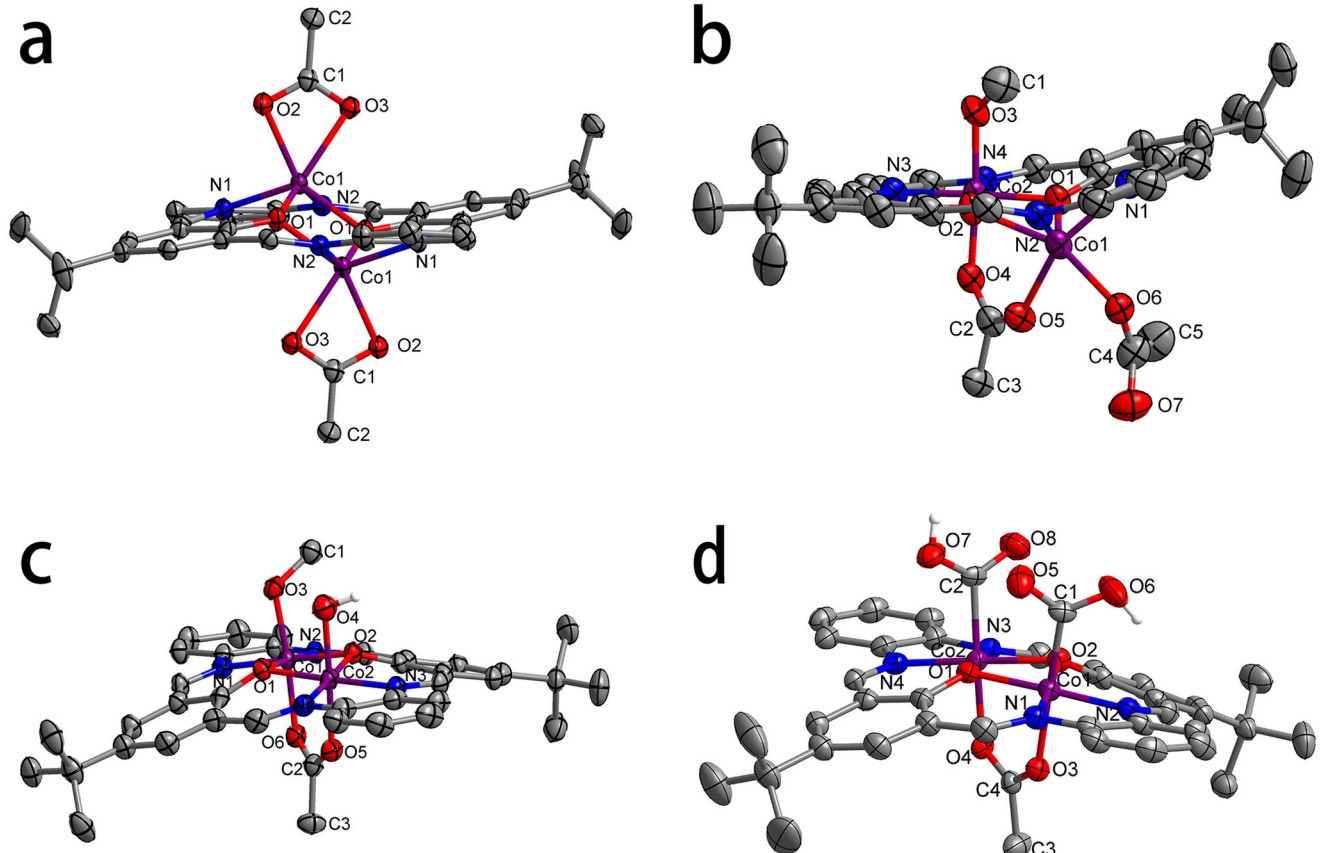

**Fig. 3 | ORTEP representations (50% probability) of the X-ray structures. a** For **1**. **b** For **2**. **c** For the cationic part of **3**. **d** For the cationic part of **4**. Crystallized solvent molecules, counteranions, and hydrogen atoms, except for those on oxygen atoms, have been omitted for clarity. For details about these X-ray structures see Supplementary Data 1.

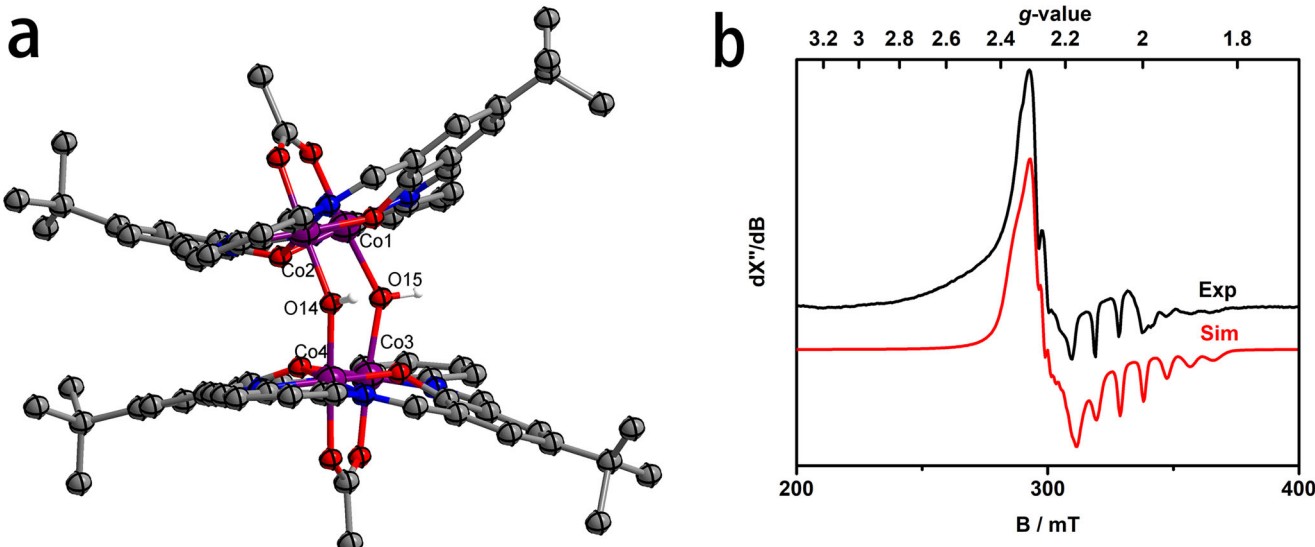

**Fig. 4 | Characterizations of 5. a** ORTEP representation (50% probability) of the cationic part of **5**. Crystallized solvent molecules, counteranions, and hydrogen atoms, except for those on oxygen atoms, have been omitted for clarity. For details about this X-ray structure see Supplementary Data 1. **b** Experimental (black) and simulated (red) X-band EPR spectrum of **5** in solid state.

production of oxalic acid as a consequence of the catalyst decomposition. *XPS* measurement of this precipitate shows the presence of unidentified Co(II) complexes (Supplementary Fig. 23). Of note, both **3** and **5** are also capable for catalyzing the production of oxalic acid under same conditions, but in lower efficiency with TONs of 10.4 and

11.3, respectively, while **1** and **2** are not suitable candidates for this catalysis (Table 1). To further confirm the origins of the carbonyl and hydroxyl groups in the oxalic acid product, isotope labelling experiments have been conducted. In $^{13}$C-labelling experiment, the produced oxalic acid is converted to calcium oxalate by reacting with $CaCl_2$ and

**Table. 1 | The efficiency and selectivity of different catalysts for oxalic acid production**

| TONs | | | | | | | |
|---|---|---|---|---|---|---|---|
| **Catalyst** | (HO–CO–CO–OH) | (MeO–CO–CO–OMe) | (MeO–CO–OMe) | (O=CH–O–Me) | (O=CH–OH) | **$CO_2$** | **$H_2O_2$** |
| **1** | n.d. | n.d. | 0.2 | n.d. | trace | n.d. | n.d. |
| **2** | 0.4 | n.d. | 0.2 | n.d. | trace | n.d. | n.d. |
| **3** | 10.4 | n.d. | 1.3 | 1.4 | trace | n.d. | 0.9 |
| **4** | 38.5 | n.d. | 0.3 | 2.4 | 0.6 | n.d. | 0.1 |
| **5** | 11.3 | n.d. | trace | 2.7 | trace | n.d. | 0.1 |

All the catalytic reactions were carried out in methanol under 2 atm of $CO/O_2$ mixture gas (1:1 v/v) at room temperature under Xe-lamp irradiation for a reaction time of 28 hours; TONs were determined by $^1H$ NMR (for formic acid), gas chromatograph (for dimethyl oxalate, dimethyl carbonate, methyl formate and $CO_2$), liquid chromatograph (for oxalic acid), and chemical analysis (iodometric method, Neocuproine/$CuSO_4$ titration method and $Ce(SO_4)_2$ titration method for $H_2O_2$); n.d. not detected.

then collected for IR measurements. When $^{13}CO$ is used, the recorded C=O, C(O)-OH and (O)C-C(O) bond stretching frequencies of the obtained calcium oxalate are all red-shifted, confirming the formation of $Ca^{13}C_2O_4$ (Supplementary Fig. 24). Similar shifting trend has also been reported for $^{13}C$-labeled and unlabeled oxalates of sodium and potassium[45,46]. For the $^{18}O$-labelling experiment in which $H_2^{18}O$ and $^{16}O_2$ are used, only the $^{18}OH$ labelled oxalic acid is observed by MS measurement (m/z = 93.0 [M·H]$^-$, Supplementary Fig. 25b). In comparison, no $^{18}OH$ labeling product is seen when $H_2^{16}O$ and $^{18}O_2$ were submitted to the catalytic system (Supplementary Fig. 25c). These results support that $H_2O$ rather than $O_2$ is the source of the hydroxyl group in the produced oxalic acid.

### Computational studies on the production of oxalic acid

Given all the aforementioned results, the plausible mechanism for the oxalic acid production catalyzed by **4** is shown in Fig. 5. In addition, computational studies of all these proposed pathways based on DFT calculations have been conducted using ORCA program package[47,48] (Fig. 6). The classical migratory CO insertions into the Co(III)-OH bonds in the dinuclear hydroxocobalt(III) intermediate is principally a plausible pathway for the generation of **4**. The calculated penitential energy surface (PES) of the migratory insertion-based pathway is shown in Fig. 6a. The coordination of CO on the Co(III) center in **IN 1** is significantly endothermic with a Gibbs free energy change of 26.7 kcal/mol. The -COOH group is formed with a Gibbs free energy change (ΔG) of − 50.9 kcal/mol via the succeeding migratory CO insertion, of which a very low activation energy ($E_a$ = 0.2 kcal/mol) is calculated. For the coordination of CO to the Co(III) center in **IN 3**, the Gibbs free energy raises by 37.3 kcal/mol. Although the following migratory CO insertion only needs to traverse a small activation energy ($E_a$ = 3.0 kcal/mol) and is exothermic (ΔG = − 62.4 kcal/mol), the whole pathway is not likely to occur at ambient reaction temperature as a result of the involvements of the energetically unfavored CO coordination processes. As an alternative, a light promoted pathway for the generation of **4**, in which the Co(III)-OH bonds are photocleaved prior to the reaction with CO, is proposed (Supplementary Fig. 30 and Fig. 6b). For the first CO insertion, the coordination of CO to the Co(II) center in **IN 5** is moderately exothermic (ΔG = − 10.1 kcal/mol) and the succeeding hydroxyl radical attack on the coordinated CO that completing the formation of the first -COOH group is strongly energetically favored (ΔG = − 61.1 kcal/mol). It is worth noting that the coupling of Co(II) center with hydroxycarbonyl radical, which is formed via the feasible reaction of hydroxyl radical with CO (ΔG = −35.2 kcal/mol), is also a viable pathway for the CO insertion. A very similar PES is calculated for the second CO insertion. The calculated PESs are supportive of the smooth proceeding of the formation of **4** via the light-promoted pathway.

For the production of oxalic acid from **4**, three different pathways (Fig. 5) can be envisioned as: (i) direct -COOH group coupling in a "bimetallic" reduction elimination manner; (ii) one of Co(III)-COOH bonds in **4** is photocleaved and the succeeding attack on the intact Co(III)-COOH bond by the hydroxycarbonyl radical that yields the oxalic acid; and (iii) the photocleavage occurs for both of the Co(III)-COOH bonds in **4** and oxalic acid forms by coupling of the produced free hydroxycarbonyl radicals. The first two pathways are further investigated computationally. Very high activation energies ($E_a$ = 55.1 kcal/mol for the formation of s-E-oxalic acid and $E_a$ = 54.2 kcal/mol for the formation of s-Z-oxalic acid) are found for the direct coupling of -COOH group thus excluding their occurrences at ambient reaction temperature (Fig. 6c). For the hydroxycarbonyl radical attack based pathway, the calculated Gibbs free energy profiles show that all the radical attack processes with different multiplicities (S = 0 and 1) and hydrogen bonding patterns (O-H···O = C-OH and O-H···(OH)C = O, which lead to the formation of s-E-oxalic acid and s-Z-oxalic acid, respectively) only need to go across small energy barriers (7−9 kcal/mol) and are strongly energetically favored with large Gibbs free energy change of 32−34 kcal/mol (Fig. 6d and Supplementary Fig. 31). Therefore, the two light promoted pathways, initiated by single and double photocleavage of the Co(III)-COOH bonds in **4**, are both regarded to be responsible for the presented production of oxalic acid catalyzed by **4**.

## Discussion

In summary, a series of dicobalt complexes (**1-4**) bearing planar macrocyclic ligands have been synthesized and characterized. The light irradiation of the rare coplanar dinuclear hydroxycarbonylcobalt(III) complex (**4**) leads to the photocleavage of the Co(III)-COOH bonds and the formation of oxalic acid. Inspired by this result, a strategy for the direct and selective production of oxalic acid with good atom economy via oxidative CO coupling at ambient conditions mediated by **4** is developed. The $^{13}C$- and $^{18}O$-labelling experiments confirm that CO and $H_2O$ are the sources of the -COOH groups in **4** and oxalic acid product. The presented results may provide the basis for developing new strategy for CO upgrading, and shed light in designing of bimetallic complex platforms with novel reactivity. Further explorations on extending the reactivity scope of this bimetallic system are underway.

## Methods

All manipulations involving air-sensitive materials were performed under $N_2$ atmosphere using standard Schlenk techniques or in gloveboxes. Chemicals were purchased from Sigma-Aldrich, Alfa Aesar, J&K Scientific Ltd. or Cambridge Isotope Laboratory Inc. All chemicals were used without further treatment.

### Synthesis of ligand and dicobalt complexes

**H₂L.** To a stirred solution of 5-t-butyl-2-hydroxyisophthalaldehyde (0.7828 g, 3.800 mmol) in methanol (40.0 mL) containing AcOH (1840 μL, 31.04 mmol) was added a solution of o-phenylenediamine

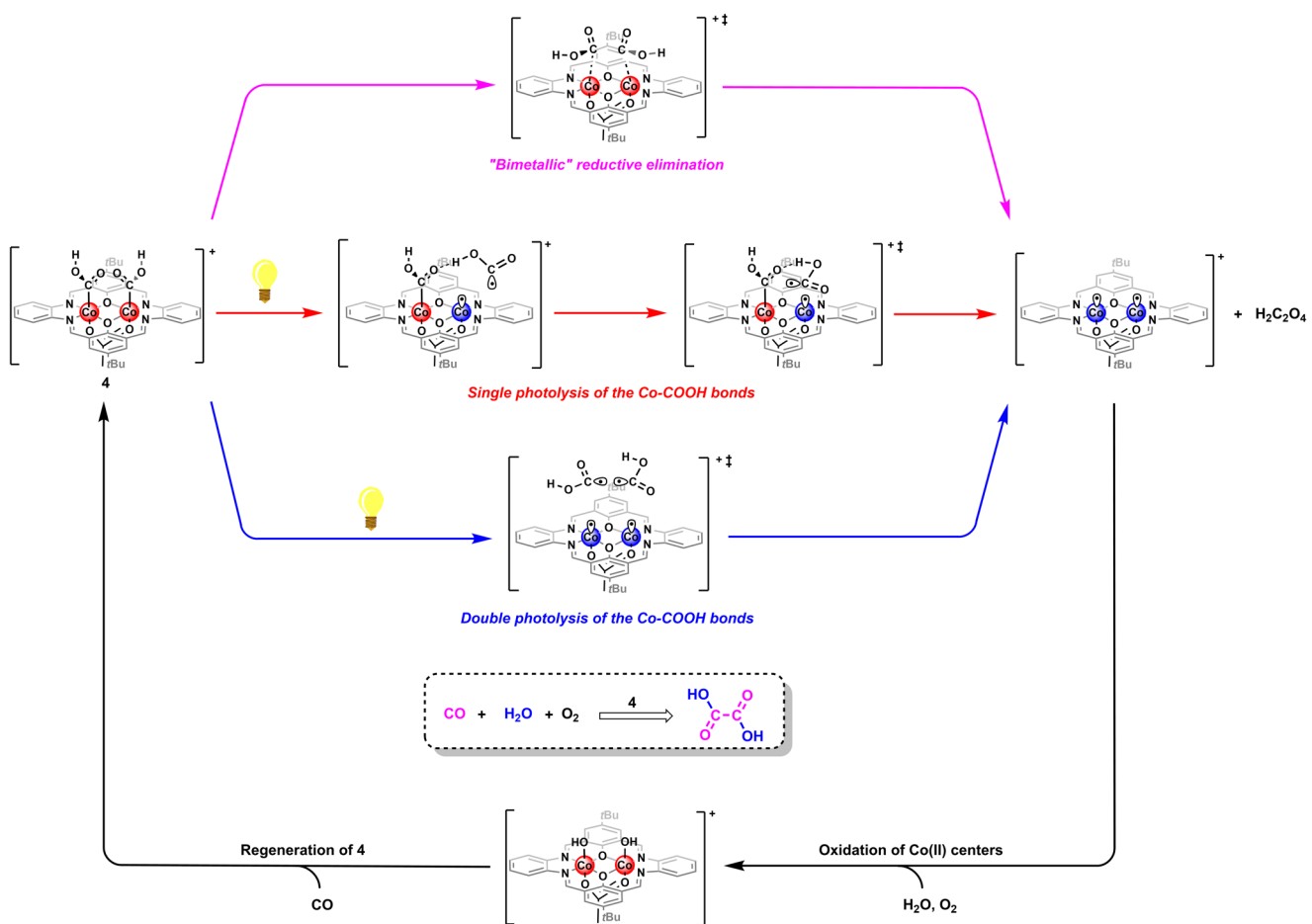

**Fig. 5 | Plausible mechanism for the production of oxalic acid catalyzed by 4.** The counteranions have been omitted for clarity.

(0.4104 g, 3.800 mmol) in methanol (40.0 mL) dropwise for 30 min at room temperature and then stirred for 6 h. The methanol solution was then removed by filtration, and the residual solid was collected, followed by the removal of all volatiles under vacuum, giving the title ligand as a yellow powder (56% yield). $^1$**H NMR** (400 MHz, CDCl$_3$) $\delta$ (ppm): 13.55 (s, 2H, O*H*), 8.63 (s, 2H, NC*H*), 7.42 (s, 1H, Ar*H*), 7.42 (s, 1H, Ar*H*), 7.35 (s, 1H, Ar*H*), 7.34 (s, 1H, Ar*H*), 7.07 (d, *J* = 7.72 Hz, 1H, Ar*H*), 7.05 (d, *J* = 7.72 Hz, 1H, Ar*H*), 6.98 (br, 1H, Ar*H*), 6.96 (br, 1H, Ar*H*), 6.79 (t, *J* = 7.56 Hz, 1H, Ar*H*), 6.78 (t, *J* = 7.44 Hz, 1H, Ar*H*), 6.32 (t, *J* = 5.60 Hz, 2H, Ar*H*), 4.46 (s, 2H, Ar*H*), 4.45 (s, 2H, Ar*H*), 3.50 (s, 1H, CN*H*), 3.49 (s, 1H, CN*H*), 1.33 (s, 18H, *t*Bu*H*).

**(L´)(AcO)$_2$Co(II)$_2$ (1).** In a nitrogen-filled glovebox, H$_2$L (0.0589 g, 0.105 mmol) was dissolved in 8.0 mL of ethanol and followed by the addition of Co(OAc)$_2$·4H$_2$O (0.0549 g, 0.220 mmol). The reaction mixture was stirred for 6 h at 85 °C, during which time the solution developed from yellow suspension to brown solution. The obtained solution was concentrated to 1.0 mL under reduced pressure, and diethyl ether (30.0 mL) was added. The precipitate was collected by filtration and then dried under vacuum to afford **1** in 86% yield. Crystals suitable for X-ray diffraction were grown by slow evaporation of ethanol solution of **1** at 1 °C for 24 h. **UV/Vis** (ethanol): 304 nm ($\varepsilon$ = 5.53 × 10$^3$ L mol$^{-1}$ cm$^{-1}$), 314 nm ($\varepsilon$ = 5.54 × 10$^3$ L mol$^{-1}$ cm$^{-1}$), 331 nm ($\varepsilon$ = 3.61 × 10$^3$ L mol$^{-1}$ cm$^{-1}$), 412 nm ($\varepsilon$ = 2.69 × 10$^3$ L mol$^{-1}$ cm$^{-1}$). **Analysis** (calcd., found for C$_{40}$H$_{40}$Co$_2$N$_4$O$_6$·3H$_2$O): C (56.88, 56.81), H (5.49, 5.51), N (6.63, 6.43).

**(L´)(AcO)$_2$(MeO)Co(II)Co(III) (2).** Complex **1** (0.0844 g, 0.100 mmol) was charged into a 25.0 mL Schlenk flask, followed by the addition of

methanol (6.0 mL) at room temperature. After three freeze-pump-thaw cycles, 1 atm of O$_2$ was inflated into the Schlenk flask. The solution was then stirred for 30 min, yielding a red solution. The obtained solution was concentrated to 1.0 mL under reduced pressure, and diethyl ether (10.0 mL) was added. The orange precipitate was collected by filtration and then dried under vacuum to afford **2** in 81% yield. X-ray quality crystals of **2** could be obtained from a concentrated methanol solution over three days at room temperature. **UV/Vis** (ethanol): 298 nm ($\varepsilon$ = 7.43 × 10$^3$ L mol$^{-1}$ cm$^{-1}$), 418 nm ($\varepsilon$ = 2.13 × 10$^3$ L mol$^{-1}$ cm$^{-1}$), 474 nm ($\varepsilon$ = 1.25 × 10$^3$ L mol$^{-1}$ cm$^{-1}$). **Analysis** (calcd., found for C$_{41}$H$_{44}$Co$_2$N$_4$O$_7$·H$_2$O): C (58.58, 58.13), H (5.52, 5.46), N (6.66, 6.39).

**[(L´)(AcO)(MeO)(OH)Co(III)$_2$]$_2$[Ce(NO$_3$)$_6$] (3).** To a solution of complex **2** (0.1210 g, 0.1440 mmol) in 8.0 mL methanol was added Ce(NH$_4$)$_2$(NO$_3$)$_6$ (0.0868 g, 0.158 mmol). After three freeze-pump-thaw cycles, 1 atm of O$_2$ was inflated into a 25.0 mL Schlenk flask. The reaction mixture was stirred for 1 h at 80 °C resulting in a color change from red to brown. The solution was concentrated to 4.0 mL under reduced pressure, and diethyl ether (10.0 mL) was added. The brown precipitate was collected by filtration and then dried under vacuum to afforded **3** in 86% yield. X-ray quality crystals of **3** could be obtained from a concentrated methanol solution over three days at room temperature. **UV/Vis** (ethanol): 300 nm ($\varepsilon$ = 7.02 × 10$^3$ L mol$^{-1}$ cm$^{-1}$), 331 nm ($\varepsilon$ = 7.58 × 10$^3$ L mol$^{-1}$ cm$^{-1}$), 438 nm ($\varepsilon$ = 1.60 × 10$^3$ L mol$^{-1}$ cm$^{-1}$). $^1$**H NMR** (400 MHz, CD$_3$OD) $\delta$ (ppm): 9.22 (s, 2H, NC*H*), 9.12 (br, 2H, NC*H*), 8.50–8.47 (m, 4H, Ar*H*), 8.34–8.32 (m, 4H, Ar*H*), 7.79–7.72 (m, 4H, Ar*H*), 1.66 (s, 3H, C*H*$_3$COO), 1.47–1.46 (br, 18H, *t*Bu*H*). **IR** (potassium bromide disk technique): $\nu$(O-H) = 3425 cm$^{-1}$. **Analysis** (calcd.,

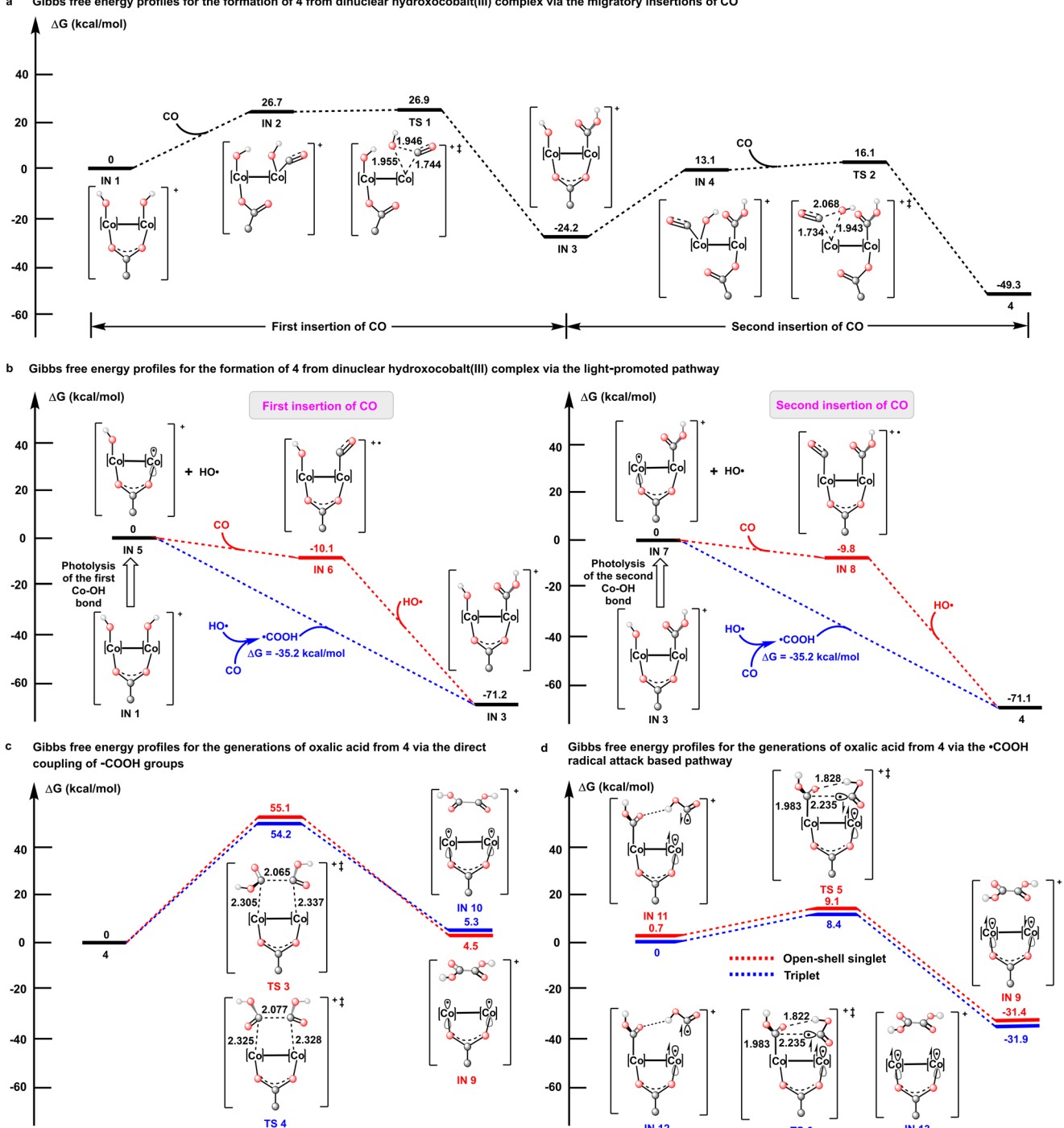

**Fig. 6 | Mechanistic investigations.** Gibbs free energy profiles for the formation of **4** from dinuclear hydroxocobalt(III) complex via: a The classical migratory insertions of CO; **b** The light-promoted pathway. Gibbs free energy profiles for the generations of oxalic acid from **4** via: **c** The direct coupling of -COOH groups; **d** The hydroxycarbonyl radical attack based pathway. Bond lengths and distances are provided in Å, detailed structures of the intermediates and transition states have been summarized in Supplementary Data 2. The counteranions have been omitted for clarity.

found for C$_{78}$H$_{82}$CeCo$_4$N$_{14}$O$_{30}$·H$_2$O): C (44.84, 44.99), H (4.05, 4.13), N (9.39, 9.82).

**[(Lʹ)(AcO)(COOH)$_2$Co(III)$_2$][NO$_3$] (4).** This complex could be synthesized by two different methods. Method A: In a 25.0 mL Schlenk flask, a solution of complex **3** (0.0350 g, 0.0170 mmol) in 4.0 mL methanol was degassed by three freeze-pump-thaw cycles. 1 atm of O$_2$ was inflated into the Schlenk flask, followed by the addition of 1 atm of CO. The solution was stirred for 24 h at 50 °C. The red precipitate was

collected by filtration to give **4** as an analytically pure dark red powder in 26% yield. Crystals suitable for X-ray diffraction were obtained by slow evaporation of methanol solution of **4** at room temperature. Method B: In a 25.0 mL Schlenk flask, a solution of complex **5** (0.0320 g, 0.00800 mmol) in 4.0 mL methanol was degassed by three freeze-pump-thaw cycles. 1 atm of O$_2$ was inflated into the Schlenk flask, followed by the addition of 1 atm of CO. The solution was stirred for 24 h at 50 °C. The red precipitate was collected by filtration to give **4** as an analytically pure dark red powder in 39% yield. Crystals suitable

for X-ray diffraction were obtained by slow evaporation of methanol solution of **4** at room temperature. **UV/Vis** (ethanol): 304 nm ($\varepsilon = 6.73 \times 10^3$ L mol$^{-1}$ cm$^{-1}$), 335 nm ($\varepsilon = 6.21 \times 10^3$ L mol$^{-1}$ cm$^{-1}$), 354 nm ($\varepsilon = 5.74 \times 10^3$ L mol$^{-1}$ cm$^{-1}$), 369 nm ($\varepsilon = 4.95 \times 10^3$ L mol$^{-1}$ cm$^{-1}$), 445 nm ($\varepsilon = 1.76 \times 10^3$ L mol$^{-1}$ cm$^{-1}$). **IR** (potassium bromide disk technique): $\nu$(C = O) = 1697 and 1670 cm$^{-1}$; $\nu$($^{13}$C = O) = 1662 and 1651 cm$^{-1}$. **$^1$H NMR** (400 MHz, DMSO) $\delta$ (ppm): 13.34 (s, 1H, COO*H*), 13.08 (s, 1H, COO*H*), 9.49 (s, 4H, NC*H*), 8.46 (br, 8H, Ar*H*), 7.78 (s, 4H, Ar*H*), 1.60 (s, 3H, C*H$_3$*COO), 1.45 (br, 18H, *t*Bu*H*). **$^{13}$C NMR** (400 MHz, DMSO) $\delta$ (ppm) for **4-$^{13}$C**: 172.12 (s, -COOH). **Analysis** (calcd., found for C$_{40}$H$_{40}$Co$_2$N$_5$O$_{11}$·H$_2$O): C (53.28, 53.49), H (4.58, 4.67), N (7.77, 7.55).

**[(L′)$_2$(AcO)$_2$(OH)$_2$Co$_4$(III)]$_2$[NO$_3$]$_4$[Ce(NO$_3$)$_6$] (5)**. In a 25.0 mL Schlenk flask, a solution of complex **3** (0.0299 g, 0.0330 mmol) in 8.0 mL ethanol was degassed by freeze-pump-thaw cycles. One atomsphere of O$_2$ was inflated into the Schlenk flask. The solution was stirred for 6 h at 80 °C. The precipitate was filtered off and the filtrate was dried under a vacuum. The residue was washed with cold diethyl ether (−20 °C) to give a brown powder of **5** in 42% yield. Crystals suitable for X-ray diffraction were obtained by slow evaporation of ethanol solution of **5** at room temperature. **UV/Vis** (ethanol): 300 nm ($\varepsilon = 6.47 \times 10^3$ L mol$^{-1}$ cm$^{-1}$), 324 nm ($\varepsilon = 6.26 \times 10^3$ L mol$^{-1}$ cm$^{-1}$), 442 nm ($\varepsilon = 1.41 \times 10^3$ L mol$^{-1}$ cm$^{-1}$). **IR** (potassium bromide disk technique): $\nu$(O-H) = 3379 and 3209 cm$^{-1}$. **Analysis** (calcd., found for C$_{152}$H$_{152}$CeCo$_8$N$_{26}$O$_{50}$·H$_2$O): C (48.39, 47.98), H (4.09, 4.26), N (9.66, 9.58).

**General procedure for the catalytic production of oxalic acid**. A total of 4.0 mL of a methanol solution containing 0.0170 mmol of catalysts (complexes **1**–**5**) was transferred into a 25.0 mL Schlenk flask. After three freeze-pump-thaw cycles, 1 atm of O$_2$ was inflated into the Schlenk flask, followed by the addition of 1 atm of CO. The Schlenk flask was set 20.0 cm aside from a 500 W xenon lamp at 30 °C for 28 h. The precipitate was filtered off after the reaction and the filtrate was analyzed by LCMS.

**Computational details**. All calculations were performed on the ORCA quantum chemistry program package (version 5.0.3) with the B97-3c calculation setup[49]. This setup is based on the B97 GGA functional and includes D3 with a three-body contribution and a short-range bond length correction. The modified, stripped-down triple-$\zeta$ basis, def2-mTZVP[50] is used in the setup. For the relatively large tetracobalt complex **5**, the crystal structure with the optimized positions for hydrogen atoms was used for the analysis of unrestricted corresponding orbitals and spin density.

## Data availability
The data associated with this study are available within the article, supplementary information and Supplementary Data. 1 (CIFs for the X-ray structures) and 2 (coordinates of the optimized structures used in computational studies). Crystallographic data for the structures reported in this Article have been deposited at the Cambridge Crystallographic Data Centre, under deposition numbers CCDC 2209326, 2209329, 2209330, 2209332 and 2209333. Copies of the data can be obtained free of charge via https://www.ccdc.cam.ac.uk/structures/. All data are available from the corresponding author.

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

## Acknowledgements

The authors are grateful for the financial support from the Ministry of Science and Technology of the People's Republic of China (No. 2021YFA1202400) to H.F., the National Natural Science Foundation of China (No. 22271158, No. 22071122) to H.F., and the Fundamental Research Funds for the Central Universities (Nankai University) to H.F.

## Author contributions

Y.X. performed the synthesis and characterizations of all the compounds, and the studies on the catalytic production of oxalic acid; S.L. performed the EPR data analysis and simulation; H.F. directed the research and performed the computational studies. All authors co-wrote the manuscript.

## Competing interests

The authors declare no competing interests.
