## [Peer Review File · Nature Communications]

Direct synthesis of oxalic acid via oxidative CO coupling mediated by a dinuclear hydroxycarbonylcobalt(III) complexReviewers' Comments:

Reviewer #1:

Remarks to the Author:

Direct Synthesis of Oxalic Acid via Oxidative CO coupling mediated by a coplanar dinuclear hydroxycarbonylcobalt(III) complex.

Xu and Fang report the oxidative dimerization of CO / H₂O / O₂ to form oxalic acid mediated by a dicobalt complex. The paper describes first the synthetic approach to a series of dinuclear cobalt complexes supported by a dinucleating ligand, and their redox chemistry and reaction with CO/O₂ to form a diamagnetic hydroxycarbonyl complex. Irradiation of this complex (4) leads to the formation of oxalic acid. The authors go on to show that this can be achieved catalytically with high selectivity (>95%) and TONs of 38.5.

As the authors point out the reductive coupling of CO is increasingly well studied, oxidative methods for coupling are rarer and examples of catalysis in this area are important. The data and calculations largely support the authors conclusion, however in some cases data and assignments are lacking. I am supportive of publication in Nature Commun. but have some major concerns that need to be addressed before the work can be published.

Origin of Oxygen atom: What is the catalytic reaction the authors are reporting $2 \text{CO} + 2 \text{H}_2\text{O} \rightarrow \text{C}_2\text{O}_4\text{H}_2$ is given in Fig. 1 but this is not balanced as we'd expect H₂ as well. Why is O₂ required for this reaction to work? And how does it participate. Can we confirm that O₂ is in fact not the origin of the oxygen atoms in the product oxalic acid. The authors should conduct further isotope experiments with H₂[¹⁸O] / ¹⁶O₂ and H₂[¹⁶O] / ¹⁸O₂ to confirm that, as they propose, that H₂O is in fact the origin of the remaining atomic content of the product, in addition to CO (already confirmed by labelling).

DFT calculations: The DFT calculations are incomplete. The authors need to show a complete pathway for oxidative coupling from 4 for both the S = 0 and S = 1 surface. Currently the calculations in Fig. 5 use the diradical as a zero point for C-C coupling. We need to understand the energy of formation of these diradicals from 4 (and where spin-crossover potentially occurs). For the data presented what are the products? We only have line-drawings representing radical coupling and there is essentially no information on stationary points in the SI. Does this coupling step lead directly to oxalic acid, if so what is the dicobalt product, if there is a further step that facilitates decoordination from Co (i.e. reaction with H₂O or O₂) this should also be modelled. Currently, in my opinion, the computational study is not publishable.

¹³C NMR data: Spectroscopic data are incomplete for the diamagnetic compounds. ¹³C NMR data Needs to be reported for diamagnetic compounds, especially 4 where we may expect two inequivalent ¹³C resonances for the key CO₂H ligands. If the authors struggle to observe these quaternary carbons then they could collect data on the ¹³C-labelled analogue.

Electronic Structure (magnetism): The authors also need to include more data or calculations to support the assignment of the magnetism for 1-5. This could be included in the SI. In particular compound 5 is described as a ligand-centred radical with little or no explanation of the data that support this assignment. In addition for compounds 1 and 2 which are described as S₁ = 3/2 S₂ = 3/2 and S₁ = 3/2 S₂ = 1, it would be useful for the authors to describe the other electronic states that have been considered and how they have been discounted (again this could be confined to the SI). In particular, why are we assuming ferromagnetic coupling between these sites and excluding anti-ferromagnetic coupling with e.g. with higher values of S₁ and lower S₂. EPR spectroscopy and DFT calculations would help with these assignments, especially with regards to the proposed case of ligand centred rather than cobalt centered radical character. I expect the assignments may be more complicated and nuanced than the authors make out in the manuscript.

Reviewer #2:

Remarks to the Author:

Fang and co-workers reported a direct synthesis of oxalic acid via oxidative CO coupling mediated by a dinuclear cobalt(III) complex. The ¹³C-labelling experiments suggest that CO is the source of the carbonyl groups in the dinuclear Co(III) complex and the product of the oxalic acid. As it is the first time to achieve light-promoted catalytic synthesis of oxalic acid from CO and H₂O with O₂ as the oxidant, the novelty of the findings presented in this manuscript is sufficient for its publication in this journal. The following issues are suggested for the improvement of the manuscript.

1. It seems that the macrocyclic ligand is crucial for achieving the direct synthesis of oxalic acid. Did the authors try other polydentate ligands? If all failed, the authors should comment the specificity of this ligand. Could the tBu groups be replaced by other substituents?
2. Dinuclear Co carbonyl complexes were also able to achieve hyperconjugation (DOI: 10.1039/D0CP06388F), how about the aromaticity in the metal-containing five-membered rings?
3. The bonding situation in complexes 1-5 should be examined. Is there any weak metal-metal bonding in these complexes? What are the bond orders of the metal-metal bond in these complexes?
4. Why did the authors choose B97-3C density functional for the calculations? How about the performance of other density functionals, e.g., TPSS, PBE0, M06 and M06L? The reliability of the chosen density functional should be investigated. The authors could make a comparison of the optimized structures at various density functionals with the experimental data, e.g., bond distance, bond angle and dihedral angle.
5. Figure 5 only shows the Gibbs energies for the formation of oxalic acid from 4 via radical pathways. The whole reaction pathway proposed in Figure 4b should be examined by DFT calculations. The authors provided the structures of reactants and transition states in Figure 5. The structures of the products should be also provided in Figure 5 for comparison. In addition, the calculated spin density should be also provided in Figure 5.

Reviewer #3:

Remarks to the Author:

Fang and co-worker present a first study on the catalytic generation of oxalic acid from CO by a dinuclear cobalt complex. A sustainable, 'green' process generating the C₂ fragment oxalic acid from CO, O₂ and light is certainly a highly desirable process. The work is fundamental well designed and carried out (but see below), the individual complexes are characterised spectroscopically, structurally and, if appropriate, magnetically.

Considering this and the highly relevant subject of this work, I support publication in Nature Communications once the following points have been fully addressed.

1. I appreciate the authors emphasise on a comprehensive overview of the field, but I recommend shortening the introduction.
2. The manuscript itself is well written and reports the findings and conclusions largely supported by the data presented. However, a couple of comments have to be addressed:
 - First page of R&D: What is the evidence that the amine moieties were converted to imines? Why via cobalt mediated dehydrogenation?
 - Throughout the manuscript averaged bond length are discussed, that is fundamental OK, but I expect to see standard deviation on those values and a clear documentation which distances, in which complexes, have been averaged and are now compared. How does the Co-X distance compare

between different complexes?

- Some experiments have been reported in 'wet' solvents. Here it has to be more specific what wet means. Water content, other 'impurities', degassed?, ...

- Table 1: The TONs was determined differently for different products – Why? One consistent method should be used here. At best, two independent methods would have been used to determine all individual TONs.

- Iodometric detection of H₂O₂: This method is not specific; please use other – spectroscopic - indications for the formation of H₂O₂.

- One fundamental and crucial question is not addressed: What happens to the catalyst during catalysis? Can any of the reported complexes be isolated after catalysis? Can the catalyst be recharged/reused?

- The figures, in particular of molecular structures, should be improved. Please add atom labelling and make sure they are of sufficient size in the final publications. Please add all not carbon bound hydrogen atoms.

3. The experimental/SI part does - I my opinion - not fulfil the standards expect for publication:

- NMR: solvents, referencing, equipment, NMR tubes, atmosphere, procedures, all this must be reported more precisely.

- XRD: Here a more precise description is mandatory. See also below.

- This also applies to all other methods. With the data given (possibly besides GC) a reproduction of the experiments is unlikely to be possible. How were the samples prepared, inert conditions?, How the measurements have been conducted?, etc.

- Throughout the experimental part, the use of significant figures has to be revised.

- Elemental Analysis and ¹³C NMR data should be provided for all compounds possible.

4. Single crystal X-ray diffraction:

The XRD data in its present form, as supplied with the manuscript are not in a publishable form. The reported data are not consistent nor comprehensive within the individual cif files, nor with the manuscript (in particular regarding the used instrumentation, software, etc.).

- During the refinements of compounds 3 and – especially – 5 extensive restraints were used (for 5 four times more restraints than parameters!) Neither this, nor any other details, on the refinements or measurements are reported in the cif files or the manuscript/SI. Those details have to be addressed in detail.

All measurements have been conducted with Cu radiation – why? The crystal size seemed not to be an issue, according to the reported data. Considering the absorption properties of Co, Mo radiation would be highly recommended and is likely – if conducted to a reasonable resolution – to solve a good deal of the obvious disorder problems.

O and N atom bound hydrogen atoms, in particular if involved in H-bond should be if possible refined freely, this is not the case – why? Has this been attempted? Nor have the hydrogen bond been reported in the cif file (HTAB).

In conclusion, I strongly recommend premeasuring those structures with Mo radiation (or other harder radiation).

Comments from Reviewer 1:

Xu and Fang report the oxidative dimerization of CO / H₂O / O₂ to form oxalic acid mediated by a dicobalt complex. The paper describes first the synthetic approach to a series of dinuclear cobalt complexes supported by a dinucleating ligand, and their redox chemistry and reaction with CO/O₂ to form a diamagnetic hydroxycarbonyl complex. Irradiation of this complex (4) leads to the formation of oxalic acid. The authors go on to show that this can be achieved catalytically with high selectivity (>95%) and TONs of 38.5.

As the authors point out the reductive coupling of CO is increasingly well studied, oxidative methods for coupling are rarer and examples of catalysis in this area are important. The data and calculations largely support the authors conclusion, however in some cases data and assignments are lacking. I am supportive of publication in Nature Commun. but have some major concerns that need to be addressed before the work can be published.

Response: Thanks very much for your interest in our work and the kind support for the publication of our work.

Origin of Oxygen atom: What is the catalytic reaction the authors are reporting $2\text{CO} + 2\text{H}_2\text{O} \rightarrow \text{C}_2\text{O}_4\text{H}_2$ is given in Fig. 1 but this is not balanced as we'd expect H₂ as well. Why is O₂ required for this reaction to work? And how does it participate. Can we confirm that O₂ is in fact not the origin of the oxygen atoms in the product oxalic acid. The authors should conduct further isotope experiments with H₂[¹⁸O] / ¹⁶O₂ and H₂[¹⁶O] / ¹⁸O₂ to confirm that, as they propose, that H₂O is in fact the origin of the remaining atomic content of the product, in addition to CO (already confirmed by labelling).

Response: In this catalytic system, the total reaction equation for oxalic acid production is:

And if we cancel one molecule of H₂O at each side of the equation, the equation becomes:

When the reaction was performed under 1 atm of O₂, CO was catalytically converted to oxalic acid with a TON of 38.5. In the absence of O₂, only stoichiometric

(non-catalytic) production of oxalic acid (54% yield based on the amount of complex 4) was observed within 28 hours, via the light-induced homolysis of the Co(III)-COOH bonds in complex 4. Besides this, yellow solid was observed to precipitated out of the reaction solution during the production of oxalic acid catalyzed by 4. XPS measurement of this precipitate showed the presence of unidentified Co(II) complexes (Supplementary Fig. 23). These results indicated that O₂ acts as the oxidant, which is necessary for the regeneration of Co(III) catalyst, in the presented catalytic production of oxalic acid from CO.

Supplementary Fig. 23. XPS measurement of the precipitates formed during the production of oxalic acid catalyzed by 4.

To further confirm the origins of the oxygen atoms in the produced oxalic acid, the ¹⁸O-labelling experiment has been conducted. When H₂¹⁸O and ¹⁶O₂ were used, only the ¹⁸OH labelled oxalic acid was observed by MS measurement ($m/z = 93.0$ [M-H]⁻ Supplementary Fig. 25b). In comparison, no ¹⁸OH labeling was seen when H₂¹⁶O and ¹⁸O₂ were introduced in the reaction system (Supplementary Fig. 25c). These results supported that the H₂O rather than O₂ was the source of the hydroxyl oxygen in the produced oxalic acid. These results were added into the revised manuscript.

Supplementary Fig. 25. The MS measurements (negative mode) of $\text{H}_2\text{C}_2\text{O}_4$ standard reagent (a), and $\text{H}_2\text{C}_2\text{O}_4$ produced using $\text{H}_2^{18}\text{O}/^{16}\text{O}_2$ (b) and $\text{H}_2^{16}\text{O}/^{18}\text{O}_2$ (c).

DFT calculations: The DFT calculations are incomplete. The authors need to show a complete pathway for oxidative coupling from **4** for both the $S = 0$ and $S = 1$ surface. Currently the calculations in Fig. 5 use the diradical as a zero point for C–C coupling. We need to understand the energy of formation of these diradicals from **4** (and where spin-crossover potentially occurs). For the data presented what are the products? We only have line-drawings representing radical coupling and there is essentially no information on stationary points in the SI. Does this coupling step lead directly to oxalic acid, if so what is the dicobalt product, if there is a further step that facilitates decooordination from Co (i.e. reaction with H_2O or O_2) this should also be modelled. Currently, in my opinion, the computational study is not publishable.

Response: We want to thank reviewer 1 for these professional suggestions. We have conducted the calculation on the formation pathway of oxalic acid via the direct coupling of -COOH groups. The result is shown in Fig. 6c. Activation energies as large as ~ 55 kcal/mol were found for these processes thus excluding their occurrences at ambient reaction temperature. Furthermore, the triplet dinuclear hydroxycarbonylcobalt(III) complex (**4**) is calculated to be 24.1 kcal/mol higher in energy than its singlet state, thus the -COOH coupling on $S = 1$ surface is not favored in comparison to the process on $S = 0$ surface.

(c) Gibbs free energy profiles for the generations of oxalic acid from 4 via the direct coupling of -COOH groups

Fig. 6c. Gibbs free energy profiles for the generations of oxalic acid from 4 via the direct coupling of -COOH groups.

The energies for the formation of the singlet and triplet diradical species from 4 are 36.7 kcal/mol and 35.9 kcal/mol with the presence of O-H \cdots O=C-OH hydrogen bonding pattern, and 38.9 kcal/mol and 38.4 kcal/mol with the presence of O-H \cdots (OH)C=O hydrogen bonding pattern. The energy difference of these two spin states are very small (0.7 kcal/mol and 0.5 kcal/mol for different hydrogen bonding patterns). Actually, the optimized structures of the diradical species of different spin states are very close to each other (as being illustrated in the following figure), with RMSDs of 0.002 Å and 0.024 Å for structures with O-H \cdots O=C-OH and O-H \cdots (OH)C=O hydrogen bonding patterns, respectively. The related minimal energy crossing points of these diradical species with different spin states (also illustrated in the following figure) have also been located and the detailed information has been added into the supplementary information (Supplementary Fig. 33).

(a) Overlay of the optimized structures (left) of IN 11 (blue) and IN 12 (red) and the minimal energy crossing point (MECP) structure (right); (b) overlay of the optimized structures (left) of IN 14 (blue) and IN 15 (red) and the minimal energy crossing point (MECP) structure (right).

The coordinates of the calculated structures used in this research, including the stationary points and the transition states, have been summarized in a separate .xyz file and submitted as supplementary information. Both the direct -COOH coupling pathway and the radical attack based pathway would lead to the direct formation of oxalic acid. Co(II) containing complex was likely a resulting product of these processes, as indicated by the XPS measurements shown in Supplementary Fig. 23.

The content related to the computational studies in the manuscript has been revised based on these newly obtained results.

¹³C NMR data: Spectroscopic data are incomplete for the diamagnetic compounds. ¹³C NMR data needs to be reported for diamagnetic compounds, especially **4** where we may expect two inequivalent ¹³C resonances for the key CO₂H ligands. If the authors struggle to observe these quarternary carbons then they could collect data on the ¹³C-labelled analogue.

Response: ¹³C NMR spectra were provided for all of the diamagnetic compounds in the revised supplementary information. It is worth noting that the resonance of the quarternary carbon of the -COOH group ($\delta = 172.12$ ppm) could only be seen for ¹³C-labelled complex **4** (Supplementary Fig. 16).

Supplementary Fig. 9. ¹³C NMR spectrum of H₂L in CDCl₃.

Supplementary Fig. 11. ¹³C NMR spectrum of 3 in CD₃OD.

Supplementary Fig. 15. ^{13}C NMR spectrum of **4** in d_6 -DMSO. (* denotes the resonance of MeOH)

Supplementary Fig. 16. ^{13}C NMR spectrum of ^{13}C -labelled **4** in d_6 -DMSO. (* denotes the resonance of MeOH)

Electronic Structure (magnetism): The authors also need to include more data or calculations to support the assignment of the magnetism for **1-5**. This could be included in the SI. In particular compound **5** is described as a ligand-centred radical with little or no explanation of the data that support this assignment. In addition for compounds **1** and **2** which are described as $S_1 = 3/2$ $S_2 = 3/2$ and $S_1 = 3/2$ $S_2 = 1$, it

would be useful for the authors to describe the other electronic states that have been considered and how they have been discounted (again this could be confined to the SI). In particular, why are we assuming ferromagnetic coupling between these sites and excluding anti-ferromagnetic coupling with e.g. with higher values of S1 and lower S2. EPR spectroscopy and DFT calculations would help with these assignments, especially with regards to the proposed case of ligand centred rather than cobalt centered radical character. I expect the assignments may be more complicated and nuanced than the authors make out in the manuscript.

Response: We want to thank reviewer 1 for raising this point, which is certainly a quite complicated issue and requires further clarification. We have double checked the magnetic measurements of all the paramagnetic complexes. In addition, computational studies on the spin density of these complexes have been conducted. At room temperature, magnetization data of **1** reveals a value of $7.67 \mu_B$ which is indicative of the presence of 6 unpaired electrons based on the spin-only model. Therefore, the assignment of two high-spin ($S=3/2$, which is the highest possible spin state for the d^7 -electron configuration) Co(II) centers were made. The calculated spin density for complex **1** (Supplementary Fig. 4) showed that 2.58 unpaired electrons were found on each of the cobalt centers, consistent with the high spin ($S=3/2$) state for both of the Co(II) centers. Unrestricted corresponding orbital analysis of **1** is also in line with the assignment of two high spin Co(II) centers (Supplementary Fig. 4). Complex **2** showed an effective magnetic moment (μ_{eff}) of $4.12 \mu_B$ at room temperature based on the SQUID measurement (Supplementary Fig. 6), indicating the presence of 3 unpaired electrons based on the spin-only model. The spin density analysis of complex **2**, as shown in Supplementary Fig. 4, showed that 2.61 unpaired electrons were located on the Co(II) center. Meanwhile, unrestricted corresponding orbital analysis of **2** confirms the presence of a total of three unpaired electrons on the Co(II) center. Given all the aforementioned results, the assignment of a high spin ($S=3/2$) Co(II) center and a low spin ($S=0$) Co(III) center was made. We are sorry for the incorrect spin configuration assignment of **2** in the original manuscript.

The effective magnetic moment ($1.14 \mu_B$) measured for complex **5** at room temperature is indicative of the presence of only one unpaired electron. The unrestricted corresponding orbital analysis and the calculated spin density of complex **5** showed that the unpaired electron density was mainly on the cobalt centers (Supplementary Fig. 20). We also conducted EPR measurement of complex **5** in solid state at 97 K. An anisotropic signal ($g_1 = 2.023$, $g_2 = 2.222$, $g_3 = 2.305$) with well resolved hyperfine splitting from Co nucleus ($I = 7/2$, $A_1 = 264.00$ MHz, $A_2 = 64.50$ MHz, $A_3 = 60.00$ MHz) was observed (Fig. 4b).

The manuscript has been revised based on these newly obtained results.

Supplementary Fig. 4. Unrestricted corresponding orbitals (with overlap coefficient equals 0) of **1** (a) and **2** (b), and spin density plots of **1** (c) and **2** (d).

Supplementary Fig. 6. Plot of $\chi_M T$ vs T for **2**.

Fig. 4b. Experimental (black) and simulated (red) X-band EPR spectrum of **5**.

Supplementary Fig. 20. Unrestricted corresponding orbitals (with overlap coefficient equals 0) of **5** (a) and calculated spin density of **5** (b).

Comments from Reviewer 2:

Fang and co-workers reported a direct synthesis of oxalic acid via oxidative CO coupling mediated by a dinuclear cobalt(III) complex. The ^{13}C -labelling experiments suggest that CO is the source of the carbonyl groups in the dinuclear Co(III) complex

and the product of the oxalic acid. As it is the first time to achieve light-promoted catalytic synthesis of oxalic acid from CO and H₂O with O₂ as the oxidant, the novelty of the findings presented in this manuscript is sufficient for its publication in this journal. The following issues are suggested for the improvement of the manuscript.

Response: Thanks very much for reviewing our manuscript and your kind support for the publication of our work.

1. It seems that the macrocyclic ligand is crucial for achieving the direct synthesis of oxalic acid. Did the authors try other polydentate ligands? If all failed, the authors should comment the specificity of this ligand. Could the tBu groups be replaced by other substituents?

Response: Yes, we have tried a few other polydentate ligands. For example, we have tried the following ligand:

The cobalt complexes bearing this ligand showed very poor solubility in nearly all the solvents we have in our lab, such as MeOH, THF and toluene. The major reasons for choosing the ligand with tBu groups are: (i) the synthesis of this ligand is relatively easy; and (ii) its dicobalt complex showed reasonable solubility in solvents like methanol and DMSO.

We have tried to further improve the solubility of the dinuclear cobalt complex by installing more tBu groups by the following synthesis procedure:

However, only a complicated mixture was obtained.

2. Dinuclear Co carbonyl complexes were also able to achieve hyperconjugation (DOI: 10.1039/D0CP06388F), how about the aromaticity in the metal-containing five-membered rings?

Response: Thank reviewer 2 for this comments. This is indeed a very interesting issue. The aromaticity in the metal-containing five-membered rings (5MR) of complex **4** has been examined by the calculated nucleus-independent chemical shifts (NICSs). The results are listed in the following Table. A variety of different density functionals were used in the calculations of the NICSs. The NICS values were computed at the geometric centers of the 5MRs, as well as the points at 1 Å above and below the geometric centers of 5MRs. The negative values of the calculated NICSs indicated the aromaticity of these 5MRs.

	B3LYP	BP	TPSS	PBE0	M06	M06L
NICS(0)	-22.6/-20.6	-27.2/-25.5	-23.7/-22.2	-25.6/-23.4	-44.3/-42.3	-20.8/-19.5
NICS(1)	-13.5/-13.4	-13.9/-13.2	-13.7/-13.6	-13.8/-13.5	-12.9/-9.4	-12.6/-12.6
NICS(-1)	0/-0.9	-0.1/-1.0	-0.8/-1.4	0.5/-0.7	3.5/0.5	-0.9/-1.5

The hybrid B3LYP/M06/PBE0 functionals and GGA type TPSS/M06L/BP functional were used for geometry optimizations and NICS calculations, in combination with triple- ζ quality def2-mTZVP basis sets for all elements. The resolution of the identity plus chain of spheres approximation (RIJCOSX, for B3LYP/M06/PBE0) and the resolution of the identity approximation (RI, for TPSS/M06L/BP) were used to accelerate the calculations with the auxiliary basis set def2/J.

In comparison, when the cobalt centers and the axial ligands were removed from the optimized structures, the aromaticity significantly reduced as implied by the positive NICS values (listed in the following Table) calculated for the same set of

coordinates.

	B3LYP	BP	TPSS	PBE0	M06	M06L
NICS(0)	0.8/0.8	-0.1/0	0.1/0.2	0.8/0.8	1.0/1.1	0.3/0.3
NICS(1)	0.9/0.9	0.7/0.8	0.7/0.8	0.9/1.0	0.8/0.9	0.3/0.4
NICS(-1)	-1.2/-1.4	-1.9/-2.1	-1.7/-1.9	-1.1/-1.4	-1.2/-1.5	-1.7/-2.0

The hybrid B3LYP/M06/PBE0 functionals and GGA type TPSS/M06L/BP functional were used for geometry optimizations and NICS calculations, in combination with triple- ζ quality def2-mTZVP basis sets for all elements. The resolution of the identity plus chain of spheres approximation (RIJCOSX, for B3LYP/M06/PBE0) and the resolution of the identity approximation (RI, for TPSS/M06L/BP) were used to accelerate the calculations with the auxiliary basis set def2/J.

3. The bonding situation in complexes **1-5** should be examined. Is there any weak metal-metal bonding in these complexes? What are the bond orders of the metal-metal bond in these complexes?

Response: Mayer bond orders have been calculated for complexes **1-5**. No noticeable Co-Co bonding interaction with Mayer bond order larger than 0.1 was found.

4. Why did the authors choose B97-3C density functional for the calculations? How about the performance of other density functionals, e.g., TPSS, PBE0, M06 and M06L? The reliability of the chosen density functional should be investigated. The authors could make a comparison of the optimized structures at various density functionals with the experimental data, e.g., bond distance, bond angle and dihedral angle.

Response: The B97-3c calculation setup was very recently developed by Grimme and

coworkers (Brandenburg, J. G.; Bannwarth, C.; Hansen, A.; Grimme, S. *J. Chem. Phys.* **148**, 064104 (2018).), aiming at relatively low-cost but accurate energy and interaction calculations. This setup is based on the B97 GGA functional and includes D3 three-body contribution and a short-range bond length correction. The modified, stripped-down triple- ζ basis, def2-mTZVP basis set is used in this setup. In comparison, we made a comparison of the core coordination geometries of the optimized structures and the C=O stretching frequencies of complex **4** using various functionals, including B3LYP, BP, TPSS, PBE0, M06, and M06L, using the same basis set. The results have been added into the supplementary information and shown in Supplementary Table 3. We can see in the Table that the B97-3c calculation set up gave very good performance in both the geometry optimization and the frequency calculation among these selected functionals. Meanwhile, it is worth noting that the calculation based on B97-3c is the cheapest in the selected functional series.

Supplementary Table 3. Comparisons of the RMSDs of the selected bonds (labelled in blue) and calculated C=O stretching frequencies of **4** using different density functionals^a.

Functionals	IR/ cm ⁻¹	RMSD ^b / Å	Run time
Exp.	1697/1670	-	-
B97-3c	1724/1685	0.0325	14 h 57 min
B3LYP	1739/1704	0.0410	98 h 2 min
BP	1687/1646	0.0412	21 h 49 min
TPSS	1689/1649	0.0367	32 h 44 min
M06	1784/1756	0.0303	91 h 20 min
PBE0	1775/1743	0.0268	81 h 4 min
M06L	1756/1719	0.0375	23 h 39 min

^a The hybrid B3LYP/M06/PBE0 functionals and GGA type TPSS/M06L/BP functional were used for geometry optimizations, in combination with triple- ζ quality def2-mTZVP basis sets for all elements. The resolution of the identity plus chain of spheres approximation (RIJCOSX, for B3LYP/M06/PBE0) and the resolution of the identity

approximation (RI, for TPSS/M06L/BP) were used to accelerate the calculations with the auxiliary basis set def2/J;
^b the RMSDs were calculated for the selected bond lengths referring to the values of the crystal structure of complex 4.

5. Figure 5 only shows the Gibbs energies for the formation of oxalic acid from 4 via radical pathways. The whole reaction pathway proposed in Figure 4b should be examined by DFT calculations. The authors provided the structures of reactants and transition states in Figure 5. The structures of the products should be also provided in Figure 5 for comparison. In addition, the calculated spin density should be also provided in Figure 5.

Response: Thank reviewer 2 for this suggestion. More computational studies have been conducted, including the regeneration of the catalyst (Fig. 6a-b) and the proposed different pathways for the production of oxalic acid (Fig. 6c-d and Supplementary Fig. 31). The schematic presentations of the reactants, intermediates, transition states and products have all been displayed in Fig. 6 and Supplementary Fig. 31, those showing the calculated potential energy surfaces. The coordinates of these structures have been summarized in a .xyz file that has also been submitted as supplementary information. The spin density plots of the paramagnetic species involved in the calculated potential energy surfaces have been illustrated in Supplementary Fig. 32.

Fig. 6. Gibbs free energy profiles for the formation of **4** from dinuclear hydroxocobalt(III) complex via the classical migratory insertions of CO (a) and the light-promoted pathway (b); Gibbs free energy profiles for the generations of oxalic acid from **4** via the direct coupling of -COOH groups (c) and the hydroxycarbonyl radical attack based pathway (d) (bond lengths and distances are provided in Å, detailed structures of the intermediates and transition states are summarized in supplementary information).

Supplementary Fig. 31. Gibbs free energy profiles for the generations of oxalic acid from 4 via the hydroxycarbonyl radical attack based pathway (bond lengths and distances are provided in Å).

Supplementary Fig. 32. Spin density plots of IN 5-16 and TS 5-8.

Comments from Reviewer 3:

Fang and co-worker present a first study on the catalytic generation of oxalic acid from CO by a dinuclear cobalt complex. A sustainable, 'green' process generating the C2 fragment oxalic acid from CO, O₂ and light is certainly a highly desirable process. The work is fundamental well designed and carried out (but see below), the individual complexes are characterised spectroscopically, structurally and, if appropriate, magnetically.

Considering this and the highly relevant subject of this work, I support publication in Nature Communications once the following points have been fully addressed.

Response: We want to thank reviewer 3 for reviewing our manuscript and the positive comments for this research.

1. I appreciate the authors emphasise on a comprehensive overview of the field, but I recommend shortening the introduction.

Response: Thanks for this suggestion. We have shortened the introduction section, in which the development of the oxidative CO coupling is discussed more concisely.

2. The manuscript itself is well written and reports the findings and conclusions largely supported by the data presented. However, a couple of comments have to be addressed:

- First page of R&D: What is the evidence that the amine moieties were converted to imines?

Why via cobalt mediated dehydrogenation?

Response: The carbon-nitrogen bond lengths are used as the indexes to distinguish imine moieties (C=N bonds) from amine moieties (C-N bonds). Both of the imine and amine moieties are found in H₂L ligand (CCDC 2235093), with bond lengths of 1.272-1.280 Å and 1.445–1.447 Å, respectively. For the dicobalt complexes **1-5**, all the carbon-nitrogen bond lengths are in the range of 1.280-1.293 Å, which are very comparable to the bond lengths of the imine groups in H₂L ligand and typical values (1.280-1.284 Å) reported for other imine compounds (For example: Swamy, P. C.; Solel, E.; Reany, O.; Keinan, E. Synthetic Evolution of the Multifarene Cavity from Planar

Predecessors. *Chem. Eur. J.* **24**, 15319-15328 (2018).). In addition, the resonances of protons in both the amine ($-\underline{\text{C}}\text{H}_2\text{-NH-}$) and imine ($-\underline{\text{C}}\text{H}=\text{N-}$) moieties were seen in the ^1H NMR spectrum of H_2L ($\delta \sim 4.45$ ppm for the 4 protons in amine moieties and $\delta \sim 8.63$ ppm for the 2 protons in imine moieties, Supplementary Fig. 8). No $-\underline{\text{C}}\text{H}_2\text{-NH-}$ signal was observed in ^1H NMR spectra of the obtained diamagnetic dicobalt complexes (Supplementary Fig. 10 and Supplementary Fig. 14). These results indicated that the amine moieties in the free base H_2L ligand were converted to imine moieties during the metallation.

Without the addition of $\text{Co}(\text{OAc})_2$, no conversion or decomposition of the H_2L ligand was observed under identical conditions to the synthesis of dicobalt(II) complex **1**. To confirm the generation of H_2 during the synthesis of complex **1**, the reaction was performed in a sealed J. Young NMR tube in d_6 -DMSO at 80°C . The recorded ^1H NMR spectrum of the reaction solution showed a singlet resonance at $\delta = 4.35$ ppm that is characteristic for H_2 (Supplementary Fig. 1). Similar cobalt mediated dehydrogenation of amines was also known (Xu, R., Chakraborty, S., Yuan, H. & Jones, W. D. Acceptorless, Reversible dehydrogenation and hydrogenation of N-heterocycles with a cobalt pincer catalyst. *ACS Catal.* **5**, 6350-6354 (2015).).

Supplementary Fig. 1. ^1H NMR spectrum of the reaction solution for the synthesis of **1** in d_6 -DMSO.

- Throughout the manuscript averaged bond length are discussed, that is fundamental OK, but I expect to see standard deviation on those values and a clear documentation which distances, in which complexes, have been averaged and are now compared. How does the Co-X distance compare between different complexes?

Response: We have summarized the related bond lengths and angles of complex **1-5** in Supplementary Table 2.

Supplementary Table 2. Selected bond lengths (Å) and angles (deg) of the solid state structures of **1-5**

	Complex 1	Complex 2	Complex 3	Complex 4	Complex 5
Co-O _{equatorial} (Å)	1.989(1)/ 1.989(3)/ 2.027(2)/ 2.027(1)	2.063(2)/ 2.144(3)/ 1.881(3)/ 1.868(2)	1.914(2)/ 1.894(2)/ 1.896(2)/ 1.900(2)	1.897(2)/ 1.898(2)/ 1.889(2)/ 1.904(2)	1.918(3)/ 1.897(3)/ 1.905(4)/ 1.890(3)/ 1.890(4)/ 1.890(4)/ 1.883(4)/ 1.909(4)
Co-N _{equatorial} (Å)	2.057(2)/ 2.057(2)/ 2.077(2)/ 2.077(2)	2.108(3)/ 2.071(3)/ 1.867(3)/ 1.870(3)	1.859(3)/ 1.874(3)/ 1.868(3)/ 1.866(3)	1.870(3)/ 1.876(3)/ 1.871(3)/ 1.875(3)	1.863(5)/ 1.872(4)/ 1.871(4)/ 1.874(4)/ 1.871(4)/ 1.861(4)/ 1.861(5)/ 1.883(5)
Co-O _{AcO} (Å)	2.164(2)/ 2.164(2)/ 2.211(2)/ 2.211(2)	2.076(3)/ 2.045(3)/ 1.941(3)/	1.954(2)/ 1.919(2)	1.936(2)/ 1.969(2)	1.926(4)/ 1.907(4)/ 1.938(4)/ 1.894(4)
Co-O _{MeO} (Å)		1.884(3)	1.871(3)		
Co-C _{COOH} (Å)	-	-	-	1.918(3)/	-

					1.913(3)
Co-O _μ -OH (Å)	-	-	-	-	1.905(4)/ 1.909(4)/ 1.907(4)/ 1.895(4)
Co-O _{OH} (Å)	-	-	1.935(3)	-	-
C=O (Å)	-	-	-	1.193(5)/ 1.207(4)	-
C(O)-OH (Å)	-	-	-	1.231(5)/ 1.249(4)	-
O-C-O _{COOH} (deg)	-	-	-	120.5(3)/ 122.1(3)	-

- Some experiments have been reported in ‘wet’ solvents. Here it has to be more specific what wet means. Water content, other ‘impurities’, degassed?

Response: Thanks for this suggestion. “Wet” methanol refers to the A.R. grade methanol that is commercially available and not further dried before using. The water content in the methanol solution is ~ 0.01% *w/w* of water. More accurate description about the methanol solution has been used in the revised manuscript.

- Table 1: The TONs were determined differently for different products – Why? One consistent method should be used here. At best, two independent methods would have been used to determine all individual TONs.

Response: The properties of the theoretically plausible products of the catalytic process listed in Table 1 are quite different, therefore different determination methods were used in order to guarantee the accuracy and reliability of the results. For dimethyl oxalate, dimethyl carbonate and methyl formate, the determination could be conducted with very good sensitivity and accuracy using GC equipped with FID. However, FID showed extremely poor response to formic acid and oxalic acid. Consequently, ¹H NMR measurement was used as the alternative method for the

determination of formic acid. Before the measurement, the volatile formic acid was converted to sodium formate by its reaction with NaOH and the integral of the resonance at $\delta = 8.48$ ppm ($\underline{H}COOH$) in the recorded 1H NMR spectrum was used to determine the amount of the formic acid based on the CH_3COONa internal reference. For oxalic acid, both of the aforementioned methods were not suitable ($H_2C_2O_4$ showed extremely poor response to FID and effects like the exchange of the protons in $H_2C_2O_4$ hampered its reliable determination by 1H NMR measurement), thus the LC method, which is frequently used for oxalic acid determination in previous reports (for example: Yang, Y.L. et al. Aromatic ester-functionalized ionic liquid for highly efficient CO_2 electrochemical reduction to oxalic acid. *ChemSusChem*, **13**, 4900–4905 (2020)), was used. For CO_2 , the GC equipped with TCD was chosen to be the equipment used in the determination based on previous report (Kim, J.Y'. et al. Recovering carbon losses in CO_2 electrolysis using a solid electrolyte reactor. *Nat Catal.* **5**, 288–299 (2022)). For H_2O_2 , the widely used chemical determination methods, including iodometric method (Jung, O. et al. Highly active NiO photocathodes for H_2O_2 production enabled via outer-sphere electron transfer. *J. Am. Chem. Soc.*, **140**, 4079–4084 (2018)), Neocuproine/ $CuSO_4$ titration method (Huang, A.X. et al. Direct H_2O_2 synthesis, without H_2 Gas. *J. Am. Chem. Soc.*, **144**, 14548–14554 (2022)), and $Ce(SO_4)_2$ titration method (Yu, Z.Y. et al. Low-coordinated Pd site within amorphous palladium selenide for active, selective, and stable H_2O_2 electrosynthesis. *Adv. Mater.*, **35**, 2208101 (2023)), were more suitable and reliable than the methods used for other plausible products of the presented catalytic process.

- Iodometric detection of H_2O_2 : This method is not specific; please use other – spectroscopic - indications for the formation of H_2O_2 .

Response: Thanks for this comment. In addition to the iodometric method, two other methods have been used for the detection of H_2O_2 as followings:

(i) Neocuproine/ $CuSO_4$ titration based colorimetric method (Huang, A.X. et al. Direct H_2O_2 synthesis, without H_2 gas. *J. Am. Chem. Soc.*, **144**, 14548–14554 (2022)). The reaction solution was extracted with diethyl ether. The precipitate was filtered off and the filtrate was concentrated under reduced pressure. 2 mL of the colorimetric titrant (6 mM neocuproine, 4.2 mM $CuSO_4$, 25/75 (v/v) ethanol/DI water mixture) was added to the concentrated solution and then filtered by Nylon membrane. The UV-Vis spectrum of filtrate was then recorded. The absorbance at 454 nm was used to determine the H_2O_2 concentration based on the calibration curve (Supplementary Fig. 28).

Supplementary Fig. 28. H₂O₂ detection based on neocuproine/CuSO₄ titration: (a) UV-Vis spectra recorded for the titration of H₂O₂ standard samples. Arrow indicates the change of the absorption intensity with different [H₂O₂] (0, 0.004, 0.008, 0.012, 0.016, 0.020, 0.024, 0.028, 0.032, and 0.036 mM); (b) UV-Vis spectra recorded for the titrations of H₂O₂ formed in the production of oxalic acid catalyzed by **3** (red), **4** (blue) and **5** (olive).

(ii) Cerium sulfate titration based colorimetric method (Yu, Z.Y. et al. Low-coordinated Pd site within amorphous palladium selenide for active, selective, and stable H₂O₂ electrosynthesis. *Adv. Mater.*, **35**, 2208101 (2023)., $2 \text{ Ce}^{4+} + \text{H}_2\text{O}_2 \rightarrow 2 \text{ Ce}^{3+} + 2 \text{ H}^+ + \text{O}_2$). The reaction solution was extracted with diethyl ether. The precipitate was filtered off and the filtrate was concentrated under reduced pressure. 2 mL of the colorimetric titrant (Methanol solution of cerium sulfate, 6 mM) was added to the concentrated solution and then filtered by Nylon membrane. The filtrate was then analyzed by UV-Vis spectrometer. The amount of H₂O₂ can be calculated as half of the consumed Ce⁴⁺ ($1 \text{ Ce}^{4+} \approx 1/2 \text{ H}_2\text{O}_2$). The concentrations of Ce⁴⁺ before and after the reaction were determined by UV-Vis spectrometer at the wavelength of 316 nm (Supplementary Fig. 29).

Supplementary Fig. 29. H₂O₂ detection based on Cerium sulfate titration: (a) UV-Vis spectra recorded for the titration of H₂O₂ standard samples. Arrows indicate the change of the absorption intensity with different [H₂O₂] (0, 0.004, 0.008, 0.012, 0.016, 0.020, 0.024, 0.028, and 0.032 mM); (b) UV-Vis spectra recorded for the titrations of H₂O₂ formed in the production of oxalic acid catalyzed by **3** (red), **4** (blue) and **5** (olive).

These newly obtained results have been added into the revised supplementary information.

- One fundamental and crucial question is not addressed: What happens to the catalyst during catalysis? Can any of the reported complexes be isolated after catalysis? Can the catalyst be recharged/reused?

Response: Yellow solid was observed to precipitated out of the reaction solution during the catalytic production of oxalic acid. This precipitate showed very poor solubility in methanol but dissolved well in DMSO. No identified compound could be found in the ¹H NMR spectrum recorded for this precipitate (Supplementary Fig. 22). XPS measurement of this precipitate showed the presence of Co(II) containing species (Supplementary Fig. 23). Therefore, the decomposition of the catalyst (complex **4**) was observed during the catalysis process and, unfortunately, could not be reused.

Supplementary Fig. 22. ^1H NMR spectrum of precipitates formed during the production of oxalic acid catalyzed by **4** in d_6 -DMSO. (* denotes the resonance of MeOH).

Supplementary Fig. 23. XPS measurement of the precipitates formed during the production of oxalic acid catalyzed by **4**.

- The figures, in particular of molecular structures, should be improved. Please add atom labelling and make sure they are of sufficient size in the final publications. Please add all not carbon bound hydrogen atoms.

Response: We are sorry for the negligence. The atom labels and hydrogen atoms that were not bound to carbon atoms have been added in the revised manuscript.

3. The experimental/SI part does - I my opinion - not fulfil the standards expect for publication:

- NMR: solvents, referencing, equipment, NMR tubes, atmosphere, procedures, all this must be reported more precisely.

Response: Thank reviewer 3 for raising this point, which certainly requires further clarification. More detailed information, as shown below, has been added into the revised supplementary information:

Deuterated solvents (CD_3OD , DMSO and CDCl_3) were purchased from Cambridge Isotope Laboratory Inc, and used without further treatment. ^1H NMR and ^{13}C NMR spectra were recorded on a Bruker Ascend™ 400 spectrometer at 298 K, and the chemical shifts were referenced to solvent residual signals. The common glass NMR tubes with an outside diameter of 5 mm (Wilmad WG-1000) were used for diamagnetic compounds, and J Young tubes (Wilmad GVA-5, with an outside diameter of 5 mm) were used for samples for which specific atmosphere were required. For ^1H NMR measurements, 0.0020 ~ 0.0030 mg of the sample was dissolved in 0.4 mL of deuterated reagents in the NMR tube and then the measurement was conducted; For ^{13}C NMR measurements, 0.0050 ~ 0.0100 mg of the sample was dissolved in 0.4 mL of deuterated reagents in the NMR tube and then the measurement was conducted.

- XRD: Here a more precise description is mandatory. See also below.

Response: More information about the XRD data has been summarized in Supplementary Table 1.

Supplementary Table 1. Summary of crystallographic data collection and structure refinement for **1-5**

	1	2	3	4	5
Empirical formula	C ₄₈ H ₆₂ Co ₂ N ₄ O ₁₀	C ₈₄ H ₉₄ Co ₄ N ₈ O ₁₆	C ₇₈ H ₈₂ CeCo ₄ N ₁₄ O ₃₀	C ₄₂ H ₄₉ Co ₂ N ₅ O ₁₄	C ₁₅₂ H ₁₆₀ CeCo ₈ N ₂₆ O ₅₄
Formula weight	972.88	885.74	2071.42	965.72	3749.52
Crystal system	triclinic	triclinic	triclinic	monoclinic	triclinic
Space group	P-1	P-1	P-1	P21/n	P-1
T/K	100	100	100	100	100
a /Å	8.9620(4)	9.2874(4)	12.7711(2)	8.64780(10)	13.9916(4)
b /Å	9.3819(5)	16.0183(7)	13.5945(2)	29.4492(2)	17.4462(5)
c /Å	16.2138(6)	18.0697(7)	14.5111(2)	16.36320(10)	21.1759(5)
α /deg	74.788(4)	115.815(4)	76.5690(10)	90	75.589(2)
β /deg	79.401(3)	104.125(3)	80.1960(10)	94.4990(10)	79.057(2)
γ /deg	62.145(5)	90.079(3)	89.0770(10)	90	74.055(2)
V /Å ³	1160.07(11)	2328.85(19)	2413.96(6)	4154.39(6)	4772.7(2)
Z	1	2	1	4	1
D /g cm ⁻³	1.393	1.263	1.425	1.544	1.305
μ /mm ⁻¹	6.111	5.988	9.524	6.906	7.695
F(000)	512.0	926	1054.0	2008.0	1917.9
Reflections collected	10257 4689	19470 9189	23217 9851	23191 8663	50290 16653
Independent reflections	R _{int} = 0.0335 R _{sigma} = 0.0396	R _{int} = 0.0433 R _{sigma} = 0.0596	R _{int} = 0.0332 R _{sigma} = 0.0304	R _{int} = 0.0326 R _{sigma} = 0.0377	R _{int} = 0.0562 R _{sigma} = 0.0643
Data/restraints/parameters	4689/0/308	9189/0/516	9851/324/689	8663/4/582	16612/4368/1095
Goodness-of-fit on F ²	1.064	1.104	1.047	1.068	1.062
R ₁ / wR ₂ indexes [I >2 σ (I)]	R ₁ = 0.0489 wR ₂ = 0.1278	R ₁ = 0.0712 wR ₂ = 0.2027	R ₁ = 0.0579 wR ₂ = 0.1673	R ₁ = 0.0591 wR ₂ = 0.1762	R ₁ = 0.0678 wR ₂ = 0.1898
R ₁ / wR ₂ indexes [all data]	R ₁ = 0.0537 wR ₂ = 0.1306	R ₁ = 0.0796 wR ₂ = 0.2097	R ₁ = 0.0595 wR ₂ = 0.1690	R ₁ = 0.0645 wR ₂ = 0.1809	R ₁ = 0.0957 wR ₂ = 0.2080

- This also applies to all other methods. With the data given (possibly besides GC) a reproduction of the experiments is unlikely to be possible. How were the samples prepared, inert conditions? How the measurements have been conducted?, etc.

Response: Thank reviewer 3 for raising this point, which certainly requires further clarification. More information, as shown below, has been added into the revised supplementary information:

Formic acid detection: 4.0 mL of methanol solution containing 0.0170 mmol of complex **4** was transferred into a 25.0 mL Schlenk flask. After three freeze-pump-thaw cycles, 1 atm of O₂ was inflated into the Schlenk flask, followed by the addition of 1 atm of CO. The Schlenk flask was set 20.0 cm aside a 500 W xenon lamp at 30 °C for 28 h. Then, excess amount of NaOH was added to the reaction solution at room temperature in an air atmosphere, and then the reaction solution was dried and used for NMR analysis.

Oxalic acid detection: 4.0 mL of methanol solution containing 0.0170 mmol of complex **4** was transferred into a 25.0 mL Schlenk flask. After three freeze-pump-thaw cycles, 1 atm of O₂ was inflated into the Schlenk flask, followed by the addition of 1 atm of CO. The Schlenk flask was set 20.0 cm aside a 500 W xenon lamp at 30 °C for 28 h. The precipitate was filtered off and the filtrate was directly analyzed by LCMS at room temperature in an air atmosphere. (The amount of oxalic acid product was detected by LCMS, performed on Shimadzu LCMS-2020 instrument equipped with a UV detector using a Shim-Pack Scepter C18-120 reverse column (3 μm, 3×33 mm). The mobile phases were acetonitrile and water/5 mM NH₄HCO₃ solution, and the flow ratio was 5:95 at a total flow rate of 1.5 mL/min.)

CO₂ detection: 4 mL of methanol solution containing 0.0170 mmol of complex **4** was transferred into a 25.0 mL Schlenk flask. After three freeze-pump-thaw cycles, 1 atm of O₂ was inflated into the Schlenk flask, followed by the addition of 1 atm of CO. The Schlenk flask was set 20.0 cm aside a 500 W xenon lamp at 30 °C for 28 h. After the reaction, the gas is collected by balloon and then connected to GC injector for gas analysis.

Dimethyl oxalate, dimethyl carbonate and methyl formate detection: 4.0 mL of methanol solution containing 0.0170 mmol of complex **4** was transferred into a 25.0 mL Schlenk flask. After three freeze-pump-thaw cycles, 1 atm of O₂ was inflated into the Schlenk flask, followed by the addition of 1 atm of CO. The Schlenk flask was set 20.0 cm aside a 500 W xenon lamp at 30 °C for 28 h. The precipitate was filtered off and the filtrate was then analyzed by GC.

- Throughout the experimental part, the use of significant figures has to be revised.

Response: We have revised the significant figures of the experimental part in revised manuscript.

- Elemental Analysis and ^{13}C NMR data should be provided for all compounds possible.

Response: The ^{13}C NMR spectra has been provided for all of the diamagnetic compounds in the revised supplementary information. Elemental analysis results have also been provided for all compounds in the revised supplementary information.

Supplementary Fig. 9. ^{13}C NMR spectrum of H₂L in CDCl₃.

Supplementary Fig. 11. ¹³C NMR spectrum of complex 3 in CD₃OD.

Supplementary Fig. 15. ¹³C NMR spectrum of 4 in *d*₆-DMSO. (* denotes the resonance of MeOH)

Supplementary Fig. 16. ^{13}C NMR spectrum of ^{13}C -labelled **4** in d_6 -DMSO. (* denotes the resonance of MeOH)

Following listed the EA results:

	Calculated (wt%)	Experimental (wt%)
(1): $\text{C}_{40}\text{H}_{40}\text{Co}_2\text{N}_4\text{O}_6 \cdot 3\text{H}_2\text{O}$	C, 56.88; H, 5.49; N, 6.63	C, 56.81; H, 5.51; N, 6.43
(2): $\text{C}_{41}\text{H}_{44}\text{Co}_2\text{N}_4\text{O}_7 \cdot \text{H}_2\text{O}$	C, 58.58; H, 5.52; N, 6.66	C, 58.13; H, 5.46; N, 6.39
(3): $\text{C}_{78}\text{H}_{82}\text{CeCo}_4\text{N}_{14}\text{O}_{30} \cdot \text{H}_2\text{O}$	C, 44.84; H, 4.05; N, 9.39	C, 44.99; H, 4.13; N, 9.82
(4): $\text{C}_{40}\text{H}_{40}\text{Co}_2\text{N}_5\text{O}_{11} \cdot \text{H}_2\text{O}$	C, 53.28; H, 4.58; N, 7.77	C, 53.49; H, 4.67; N, 7.55
(5): $\text{C}_{152}\text{H}_{152}\text{CeCo}_8\text{N}_{26}\text{O}_{50} \cdot \text{H}_2\text{O}$	C, 48.39, H, 4.09; N, 9.66	C, 47.98; H, 4.26; N, 9.58

4. Single crystal X-ray diffraction:

The XRD data in its present form, as supplied with the manuscript are not in a publishable form. The reported data are not consistent nor comprehensive within the individual cif files, nor with the manuscript (in particular regarding the used instrumentation, software, etc.).

Response: Thanks for pointing this out. The corresponding parts in both the

supplementary information and the CIFs have been checked and corrected. Crystallographic data was collected on Rigaku XtalAB Pro MM007 DW diffractometer with graphite monochromated Cu K α radiation ($\lambda = 1.54178$ Å). Structure was solved using direct methods and then refined using SHELXL-2014 and Olex2 (Sheldrick, G. M. SHELXT – Integrated space-group and crystal-structure determination. *Acta Cryst. A.* **71**, 3–8 (2015); Dolomanov, O. V. et al. OLEX2: a complete structure solution, refinement and analysis program. *J. Appl. Crystallogr.* **42**, 339–341 (2009); Spek, A. L. Structure validation in chemical crystallography. *Acta Cryst. D.* **65**, 148–155 (2009)) to convergence, in which all the non-hydrogen atoms were refined anisotropically during the final cycles. All hydrogen atoms of the organic molecule were placed by geometrical considerations and were added to the structure factor calculation.

- During the refinements of compounds **3** and – especially – **5** extensive restraints were used (for **5** four times more restraints than parameters!) Neither this, nor any other details, on the refinements or measurements are reported in the cif files or the manuscript/SI. Those details have to be addressed in detail.

Response: Thanks for pointing this out. For **1**, **2**, **3** and **5**, we used the PLATON SQUEEZE procedure (Spek, A. L. PLATON SQUEEZE: a tool for the calculation of the disordered solvent contribution to the calculated structure factors. *Acta Crystallogr. Sect.C. Cryst. Struct. Commun.* **71**, 9–18 (2015)) to remove the uncoordinated solvent molecules which could not be modeled properly. Additionally, for **3**, we refined the structure by using some requisite restrains of anisotropy, such as RIGU and SADI for the counter cation fragments. For **4**, we refined the structure by using DELU command for the -COOH fragments. Additionally, we refined the structure by using SIMU command for **5** and omitting the some of the most disagreeable points. This detailed information has been added into the revised supplementary information.

All measurements have been conducted with Cu radiation – why? The crystal size seemed not to be an issue, according to the reported data. Considering the absorption properties of Co, Mo radiation would be highly recommended and is likely – if conducted to a reasonable resolution – to solve a good deal of the obvious disorder problems.

Response: We have tried to replace Cu radiation with Mo radiation, but the diffraction intensity is seriously weakened and the quality of the data was not as

good as that of the Cu radiation. The reason might be that the intensity of the Cu radiation is much stronger (6 times stronger) than that of the Mo radiation for our diffractometer (Rigaku XtalAB Pro MM007 DW).

O and N atom bound hydrogen atoms, in particular if involved in H-bond should be if possible refined freely, this is not the case – why? Has this been attempted? Nor have the hydrogen bond been reported in the cif file (HTAB).

Response: We have tried to locate these hydrogen atoms base on the Fo-Fc map, however no clue was found. This might be a consequence of that these H atoms orientate to many directions or even continuously around the atom they are bound to. However, the information about the hydrogen bond has been added into the revised CIFs.

Reviewers' Comments:

Reviewer #1:

Remarks to the Author:

The authors have provided a robust response and addressed the key points from the reviewers. I am supportive of publication, but would urge the correction of the equation in Figure 1. A balanced equation should be presented and the given the labelling experiments the role of O₂, as a mediator (either on both sides of the equation or above the arrow) needs to be clear as does the reaction stoichiometry.

Reviewer #2:

Remarks to the Author:

The authors have fully addressed all the concerns in the first round, thus, I believe that the revised manuscript and SI are suitable for the publication.

Reviewer #3:

Remarks to the Author:

I re-reviewed the manuscript entitled 'Direct synthesis of oxalic acid via oxidative CO coupling mediated by a dinuclear hydroxycarbonylcobalt(III) complex' by Fang and co-workers. I thank the authors for the quite comprehensive reply to my earlier remarks. I am largely satisfied with the replies. However, those replies need to find their way into the manuscript, or at least the supporting Information, e.g. the detailed reasoning for the amine-imine conversion. Moreover, in large parts the compound numbers are not consistently used., e.g. cpd. 5 is the according to Figs. 2 and 4 a cationic tetra cobalt complex, however, in the text it is mentioned that "complex 5 is obtained in 62% isolated yield" and in the experimental part 5 is defined as the compound with all its ionic. This is neither consistent nor makes it much sense. Please recheck/redefined the compound numbers carefully. The X-ray data are still not consistent between the SI and the cif files provided, e.g. with respect to the software used, measurement temperatures (cpd. 5) and in particular it is good practice to discuss the details of the refinements given in the cif files. I also suggest moving the still not comprehensive refinement details to Table 1 in the Supporting Information (rather than in the general X-ray part).

Comments from Reviewer 1:

The authors have provided a robust response and addressed the key points from the reviewers. I am supportive of publication, but would urge the correction of the equation in Figure 1. A balanced equation should be presented and the given the labelling experiments the role of O₂, as a mediator (either on both sides of the equation or above the arrow) needs to be clear as does the reaction stoichiometry.

Response: We have revised the previous equation to make it in the balanced form of reaction stoichiometry. The new Fig. 1 is as below:

a Reported synthetic routes of oxalic acid from CO

b Direct synthesis of oxalic acid via oxidative CO coupling

This work

Fig. 1. Synthetic routes of oxalic acid from CO. **a**, reported synthetic routes of oxalic acid from CO. **b**, direct synthesis of oxalic acid via oxidative CO coupling.

Comments from Reviewer 3:

I re-reviewed the manuscript entitled 'Direct synthesis of oxalic acid via oxidative CO coupling mediated by a dinuclear hydroxycarbonylcobalt(III) complex' by Fang and co-workers. I thank the authors for the quite comprehensive reply to my earlier remarks. I am largely satisfied with the replies. However, those replies need to find their way into the manuscript, or at least the supporting Information, e.g. the detailed reasoning for the amine-imine conversion. Moreover, in large parts the compound numbers are not consistently used., e.g. cpd. 5 is the according to Figs. 2 and 4 a cationic tetra cobalt complex, however, in the text it is mentioned that "complex 5 is obtained in 62% isolated yield" and in the experimental part 5 is defined as the compound with all its ionic. This is neither consistent nor makes it much sense. Please recheck/redefined the compound numbers carefully. The X-ray data are still not consistent between the SI and the cif files provided, e.g. with respect to the software used, measurement temperatures (cpd. 5) and in particular it is good practice to discuss the details of the refinements given in the cif files. I also suggest moving the still not comprehensive refinement details to Table 1 in the Supporting Information (rather than in the general X-ray part).

Response: Thank reviewer 3 very much for the reminding. The detailed reasoning for the amine-imine conversion has been added into the Supplementary Information (Page S7).

We have checked and unified the compound numbers in the main text and Supplementary Information. In the revised main text and Supplementary Information, the number label indicates the whole complex.

The details of the refinements were added into the Supplementary Information (Page S40). We double checked the information, such as the used software, measurement temperatures and diffraction measurement device type, in cif files.